# Micronutrient supplements can promote disruptive protozoan and fungal communities in the developing infant gut

Ana Popovic [1,2], Celine Bourdon[3,4], Pauline W. Wang[5,6], David S. Guttman [5,6], Sajid Soofi [7], Zulfiqar A. Bhutta[4,7], Robert H. J. Bandsma[3,4], John Parkinson [1,2,8 ✉] & Lisa G. Pell[4]

Supplementation with micronutrients, including vitamins, iron and zinc, is a key strategy to alleviate child malnutrition. However, association of gastrointestinal disorders with iron has led to ongoing debate over their administration. To better understand their impact on gut microbiota, we analyse the bacterial, protozoal, fungal and helminth communities of stool samples collected from a subset of 80 children at 12 and 24 months of age, previously enrolled into a large cluster randomized controlled trial of micronutrient supplementation in Pakistan (ClinicalTrials.gov identifier NCT00705445). We show that while bacterial diversity is reduced in supplemented children, vitamins and iron (as well as residence in a rural setting) may promote colonization with distinct protozoa and mucormycetes, whereas the addition of zinc appears to ameliorate this effect. We suggest that the risks and benefits of micronutrient interventions may depend on eukaryotic communities, potentially exacerbated by exposure to a rural setting. Larger studies are needed to evaluate the clinical significance of these findings and their impact on health outcomes.

[1] Program in Molecular Medicine, Hospital for Sick Children Research Institute, Toronto, Ontario, Canada. [2] Department of Biochemistry, University of Toronto, Toronto, Ontario, Canada. [3] Division of Gastroenterology, Hepatology and Nutrition, Hospital for Sick Children, Toronto, Ontario, Canada. [4] Centre for Global Child Health, Hospital for Sick Children, Toronto, Ontario, Canada. [5] Department of Cell & Systems Biology, University of Toronto, Toronto, Ontario, Canada. [6] Centre for the Analysis of Genome Evolution & Function, University of Toronto, Toronto, Ontario, Canada. [7] Center of Excellence in Women and Child Health, The Aga Khan University, Karachi, Pakistan. [8] Department of Molecular Genetics, University of Toronto, Toronto, Ontario, Canada. ✉email: john.parkinson@utoronto.ca

Malnutrition is a global health crisis with 149 million children stunted and 45 million children wasted under the age of 5 years[1,2]. With increased vulnerability to infection, undernourished children are at elevated risk of death, not least from diarrheal diseases[3,4]. Previous studies have demonstrated the role of gut microbiota in malnutrition, with microbiome immaturity (bacterial communities that are under-developed with respect to age) representing a key factor in disease development[5,6]. Beyond bacterial communities, parasites such as hookworm, *Cryptosporidium*, and *Entamoeba* have been associated with severe diarrheal disease and intestinal malabsorption[7,8]. However, much less is known regarding the role of other, potentially commensal, eukaryotic gut microbes in undernutrition. Of particular interest is their ability to interact with and alter bacterial communities. For example, indole-producing gut bacteria were found to confer protection against *Cryptosporidium* infection, while deworming treatments targeting helminth endemic communities reduced the abundance of pro-tective Clostridiales[9,10]. Mouse studies further showed that hel-minths and protozoa influence bacterial communities by modulating the host immune system[9,11,12]. While the number of published gut microbiome studies has increased rapidly over the last decade, few have explored the composition of eukaryotic gut communities and their potential interactions with bacteria. Pre-viously, we applied 18S rRNA and internal transcribed spacer (ITS) sequence surveys to systematically characterize eukaryotic microbiota in severely malnourished Malawian children and identified a high prevalence of protozoa including commensals and pathobionts[13]. We furthermore associated *Blastocystis* colo-nization with increased gut bacterial diversity.

Global health programs targeting vulnerable child populations include the use of micronutrient supplements, consisting of vitamins as well as essential minerals zinc and iron, that have been demonstrated to improve growth and reduce morbidity[14–16]. Such supplements are thought to address defi-ciencies that can impair immune responses to infectious patho-gens and impact gut bacterial communities[17–20]. While beneficial, supplementation, especially with iron, may also promote unin-tended growth of pathogens with the potential to disrupt the gut environment, particularly where the host is unable to restrict micronutrient bioavailability[21]. For example, it has been shown that surplus iron promotes the growth of enteropathogens and induces intestinal inflammation in infants[22,23]. Furthermore, while known to reduce the duration of childhood diarrheal epi-sodes, zinc supplementation has been associated with increased duration of *Entamoeba histolytica* infections[24,25]. The impacts of complementary foods introduced between 6 and 24 months of age on gut bacteria, and the potential of tailored formulations, often fortified with micronutrients, to alleviate undernutrition by pro-moting specific bacterial taxa were recently reviewed[26]. The success of such strategies may also depend on understanding their effect on eukaryotic microbiota and their potential to modulate the gut environment.

In this work, we attempt to understand the impact of micro-nutrient supplementation on the complex interactions between eukaryotic and bacterial microbiota in the maturing infant gut. We perform 18S rRNA and 16S rRNA amplicon surveys on stool samples obtained at 12 and 24 months of age from 80 children, previously recruited as part of a cluster randomized controlled trial (cRCT) conducted in Pakistan. The trial investigated the impact of micronutrient powders (MNP) containing vitamins and iron with or without zinc on growth and morbidity in 2746 children, and has shown an excess of significant diarrheal and dysenteric episodes among children receiving MNPs[27]. We show that diverse protozoa, helminths, and fungi colonize the gut of both undernourished children and those within a healthy weight range. Children receiving MNPs have reduced bacterial diversity, while vitamins and iron are associated with increased carriage of specific protozoa and mucormycetes, an effect mitigated by the addition of zinc. Modeling of microbiota interactions reveals a complex landscape of associations with supplementation, place of residence (i.e., urban or rural), age, and nutritional status.

## Results

**Description of cohort**. A subset of 80 children (160 paired stool samples at 12 and 24 months of age) were selected from the parent cRCT (ClinicalTrials.gov identifier NCT00705445)[27] for profiling of gut eukaryotic and bacterial communities in this study (control (n = 24), MNP (n = 29), and MNP with zinc (n = 27); Fig. 1a). The cohort includes children from both urban (Bilal colony) and rural (Matiari district) study sites. Children were stratified by weight-for-length z-scores (WLZ) at 24 months into two groups: a reference WLZ group with WLZ > −1 (mean 0.2, 95% CI [−0.1, 0.4]) and an undernourished group with WLZ < −2 (mean −2.9, 95% [CI −3.2, −2.7]). Participant characteristics are summarized in Table 1 and Supplementary Table 1, and details of subject selection are outlined in Supple-mentary Fig. 1. For comparison, summary characteristics of all participants in the parent trial followed to 24 months of age are included in Supplementary Table 1. The WLZ growth trajectories of the children selected for microbiome profiling as the reference

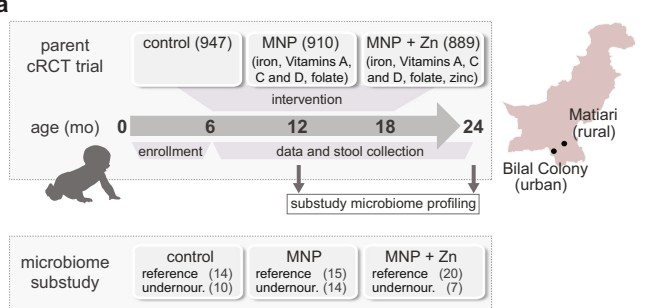

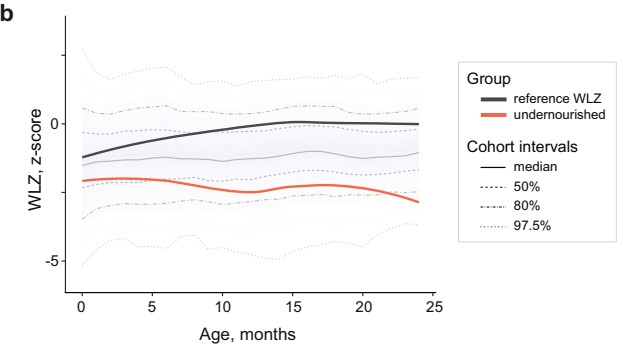

**Fig. 1 Schematic of the parent micronutrient supplementation trial and participant characteristics selected for microbiome profiling. a** The timeline of the parent cRCT (ClinicalTrials.gov identifier NCT00705445), including enrollment, intervention, and follow-up. Urban and rural study sites are shown on the right. Numbers of participants recruited into each of the three intervention arms of the parent trial are indicated in brackets. A subset of participants, undernourished (WLZ < −2) and a reference comparator group (WLZ > −1), were selected from each arm for retrospective microbiome profiling, as shown. See Supplementary Fig. 1 for details of participant selection. **b** Weight-for-length z-scores of children recruited into clinical trial NCT00705445 during the first 24 months of life. Median and quantile values are shown, with medians for participants profiled in the current study indicated by red (undernourished) and black (reference WLZ) lines.

**Table 1 Participant characteristics.**

| | Undernourished (n = 31) | Reference WLZ (n = 49) | Total (N = 80) |
|---|---|---|---|
| Rural site, n(%) | 25 (80.6%) | 28 (57.1%) | 53 (66.2%) |
| *Treatment arm, n(%)* | | | |
| Control | 10 (32.3%) | 14 (28.6%) | 24 (30.0%) |
| MNP | 14 (45.2%) | 15 (30.6%) | 29 (36.2%) |
| MNP with zinc | 7 (22.6%) | 20 (40.8%) | 27 (33.8%) |
| Female, n(%) | 14 (45.2%) | 30 (61.2%) | 44 (55.0%) |
| Premature birth, n(%) | 6 (19.4%) | 10 (20.4%) | 16 (20.0%) |
| Initiated breastfeeding, n(%) | 31 (100.0%) | 47 (95.9%) | 78 (97.5%) |
| *Breastfeeding status at 6 mo, n(%)* | | | |
| Exclusive | 0 (0%) | 0 (0%) | 0 (0%) |
| Partial | 30 (96.8%) | 48 (98.0%) | 78 (97.5%) |
| None | 1 (3.2%) | 1 (2.0%) | 2 (2.5%) |
| *Anthropometry, 12 mo* | | | |
| Weight, Kg | 6.6 (6.3, 7.0) | 8.5 (8.2, 8.8) | 7.8 (7.5, 8.1) |
| Length, cm | 69.2 (67.7, 70.7) | 71.2 (70.4, 72.1) | 70.6 (69.9, 71.4) |
| Weight-for-length, z-score | −2.4 (−3.1, −1.7) | −0.0 (−0.3, 0.2) | −0.7 (−1.1, −0.4) |
| *Anthropometry, 24 mo* | | | |
| Weight, Kg | 8.0 (7.6, 8.3) | 10.5 (10.2, 10.9) | 9.5 (9.2, 9.9) |
| Length, cm | 78.9 (77.3, 80.4) | 80.5 (79.5, 81.4) | 79.8 (79.0, 80.7) |
| Weight-for-length, z-score | −2.9 (−3.2, −2.7) | 0.2 (−0.1, 0.4) | −1.0 (−1.4, −0.7) |

Categorical values are presented as n(%), continuous variables show the mean and 95% confidence intervals. Premature birth was defined as gestational age < 37 months. Initiation of breastfeeding was reported for the period prior to recruitment into the study.

WLZ group approximately tracked the upper 50th percentile of the original cohort, while the undernourished group started around the lower 50th percentile and gradually dropped over time ending at the bottom 80th percentile of the cohort (Fig. 1b). This drop in the WLZ of the undernourished children was driven by poor weight gain (Supplementary Fig. 2). While diarrheal episodes were a primary outcome of the parent trial, with an incidence between 3.73 and 4.32 per child year during the intervention period (2.80 and 3.13 after the intervention had ceased)[27], we excluded children with reported diarrhea prior to stool collection for the purpose of microbiome analysis.

**The developing infant gut is colonized by complex eukaryotic communities.** To investigate the carriage of eukaryotic organisms within the developing infant gut, we applied 18S rRNA amplicon sequencing to all 160 stool samples. We generated a total of 11,639,233 paired amplicon reads (median 70,642) of which 7,167,464 were high-quality reads and clustered into OTUs, and 4,386,494 could be classified as a eukaryotic microbe (median 22,932; Supplementary Data 1). The remaining reads were assigned as plant (Embryophyta, 1,600,136), mammalian (312,235), arthropod (211,284), bacterial (322,725), or remained unassigned (433,435). We identified a total of 859 microbial eukaryotic OTUs (median 66; Supplementary Data 2), which included 438 protozoan, three helminth, and 418 fungal OTUs (Fig. 2a). Fungi, dominated by Mucoromycota and Ascomycota, accounted for 71% of all reads. The most abundant were species in the *Candida-Lodderomyces* clade, *Saccharomyces*, and taxa increasingly associated with rare but fatal infections known as mucormycoses: *Rhizomucor*, *Actinomucor*, and *Lichtheimia*. Alveolates accounted for 25% of reads, with *Gregarina/Gregarinasina* and *Cryptosporidium* as the most abundant (5 and 3%, respectively). Remaining reads were classified to numerous taxa, including known gut parasites such as *Enterocytozoon bieneusi*, *Pentatrichomonas hominis,* and the tapeworm *Hymenolepis nana* as well as uncharacterized alveolates, Amoebozoa, and Cercozoa (Supplementary Data 2).

Protozoa were highly prevalent, with 89% of children colonized by at least one protozoan organism by 12 months of age and 92%

by 24 months of age (Fig. 2b). Carriage of multiple species was common in both the reference WLZ and undernourished groups, with on average 18 and 19 OTUs per child at each time point and a maximum of 91. A high detection of gregarines, typically considered parasites of invertebrates, has not previously been reported in the human gut. In our cohort, gregarine sequences accounted for nearly 230,000 reads and were identified in 69 and 71% of children at 12 and 24 months of age (Fig. 2b).

**Micronutrient supplementation without zinc is associated with increased carriage of protozoa and mucormycetes.** Having generated profiles of eukaryotic communities within each sample, we were interested in identifying any links between these communities and variables captured in the parent study. Applying generalized linear models (GLMs), we tested for associations between protozoan and fungal richness (number of OTUs) and four explanatory variables: micronutrient supplementation, nutritional status, age, and rural versus urban residence. Children residing in the rural study site had increased protozoan richness compared to those from the urban setting (GLM, β = 11, 95% CI [5.3, 16.6], $p < 0.001$; Fig. 2c). Differences were attributed to the higher carriage of predominantly alveolate taxa, particularly *Cryptosporidium* (Fisher's Exact, OR 4.9, 95% CI [2, 11], $p < 0.01$), species known to cause enteric symptoms (Fig. 2d). When stratifying by age group, only *Cryptosporidium* and two OTUs classified as unknown Conoidasida, with 93% sequence identity to *Cryptosporidium*, reached statistical significance at 24 months, with 2.4 and 9.6-fold higher carriage, respectively, in children from rural settings (Fisher's Exact, OR 8.1, 95% CI [2.5, 29], $p < 0.05$, OR 15.2, 95% CI [2, 670], $p < 0.05$).

While we observed trends in increased fungal and protozoan richness in the undernourished cohort (Fig. 2c), only the tapeworm *Hymenolepis nana* was detected with overall significantly higher frequency in undernourished children (Fisher's Exact, OR 6.2, 95% CI [2, 23], $p < 0.05$; Fig. 2d). At 12 months, *H. nana* detections were only 2 and 3% in reference WLZ and undernourished children, respectively. However, by 24 months, carriage increased to 8% in reference WLZ and 43% in the undernourished group (*ns* after multiple testing correction). We also observed trends of increased carriage of *Cryptosporidium* and

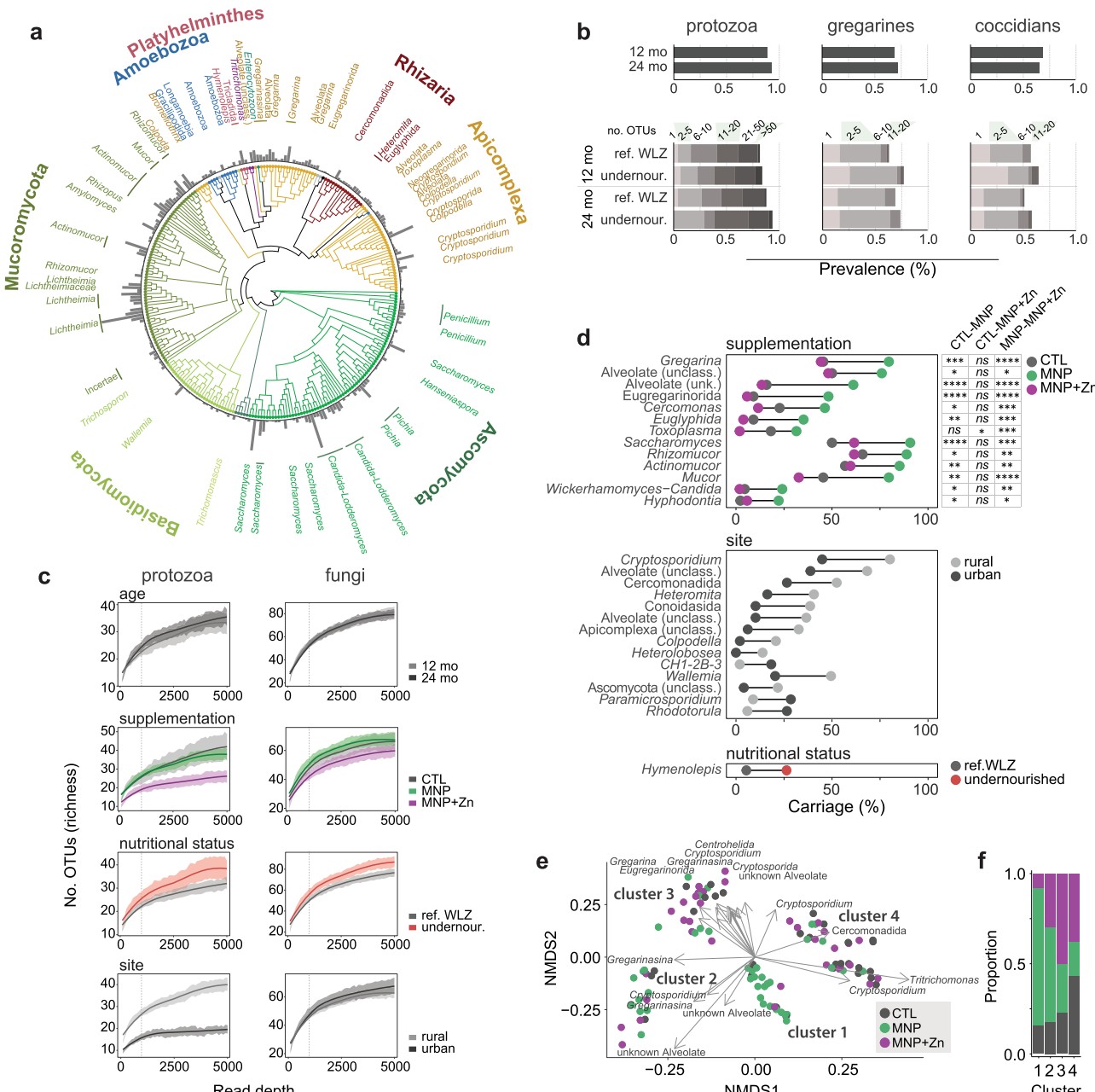

**Fig. 2 Eukaryotic communities in the gut are diverse and impacted by micronutrient supplementation and place of residence. a** Phylogenetic tree representing eukaryotic taxa detected in children. Branches are colored by phylum, and bars represent the prevalences of OTUs in the cohort. Named organisms represent those detected in more than 5% of samples with a minimum of 100 reads. **b** Prevalences of protozoan (left), and specifically gregarine (middle) or coccidian (right) OTUs detected in children at 12 and 24 months of age. Prevalences are subdivided by nutritional group in bottom graphs, where shaded regions denote binned numbers of OTUs identified per sample. **c** Rarefaction curves comparing the mean protozoan and fungal species richness by age group, micronutrient supplementation, nutritional status, and place of residence (site). Shaded regions represent standard error. Dashed lines denote the read depth at which significance was tested. **d** Carriage of eukaryotic taxa significantly associated with micronutrient supplementation, place of residence (site), or nutritional status ($n = 150$). Significant pairwise differences among supplementation groups are indicated to the right (*$p < 0.05$, **$p < 0.01$, ***$p < 0.001$, ****$p < 0.0001$, ns no significant difference) based on two-sided pairwise Fisher's Exact tests and corrected for multiple testing using the Benjamini–Hochberg method. Exact $p$-values are provided in Supplementary Data 3. **e** Non-metric multidimensional scaling of sample dissimilarities ($n = 105$) based on protozoan composition, calculated using unweighted UniFrac scores. Samples are colored by the supplementation arm, and arrows indicate the direction of increasing carriage of taxa significantly correlated with the first two ordination axes. Arrow lengths are scaled by the root square ($r^2$) of the correlation. Identified clusters are numbered 1 through 4. **f** Proportions of samples from the respective supplementation arms within each protozoan community cluster.

*Cryptosporida* (coccidians), represented by 46 OTUs in total, in undernourished children (74% versus 65% at 12 months, *ns*; 71% versus 61% at 24 months, *ns*; Fig. 2b). Furthermore, undernourished children receiving MNPs with zinc had significantly fewer protozoan OTUs relative to undernourished children in the

control and MNP arms (GLM, $\beta = -15.19$, 95% CI [$-29.27$, $-1.12$], $p < 0.05$), suggesting a possible inhibitory effect by the metal (Fig. 2c, Supplementary Fig. 3).

We next compared the phylogenetic compositions of protozoan and fungal populations. The fungal composition was not

associated with any of the variables studied here, however, analysis of protozoan compositional differences, based on unweighted UniFrac scores, revealed four distinct clusters of samples (Fig. 2e). Differences were significantly explained by place of residence (adonis, $R^2 = 0.02$, $p < 0.05$) and micronutrient supplementation (adonis, $R^2 = 0.09$, $p < 0.001$), where protozoan communities in children supplemented with MNPs differed significantly from those in control and MNPs with zinc arms (MNP vs. CTL, $R^2 = 0.05$, $p < 0.01$; MNP vs. MNP with zinc $R^2 = 0.04$, $p < 0.01$). Cluster 1 in particular was enriched in MNP samples, $X^2$ (6, $n = 105$) = 26.80, $p < 0.001$ (Fig. 2f). Key drivers of compositional differences included *Tritrichomonas*, detected exclusively in samples within clusters 1 and 4 (correlation coefficient $R^2 = 0.21$, $p = 0.001$), and an OTU assigned to an unknown alveolate found exclusively in clusters 1 and 2 ($R^2 = 0.24$, $p = 0.001$). These organisms were highly prevalent in both age groups, at 42 and 45% (*Tritrichomonas*) and 20 and 21% (unknown alveolate).

To better understand this association between protozoan community structure and micronutrient treatment, we systematically compared protozoan, as well as fungal, carriage among supplementation groups. We identified significantly higher carriages of seven phylogenetically distinct protozoa and six fungi in children receiving MNPs without zinc, relative to those that were given zinc (six protozoa and six fungi relative to the control group; Fig. 2d). Indeed, we noted a trend where MNPs with zinc reduced carriage of microbial eukaryotes to or below that observed in the control samples. For example, *Gregarina* and an uncharacterized alveolate, which contributed to the previously observed differences in beta diversity (Fig. 2e), were detected with 1.8 and 3.8-fold higher frequency in the MNP group ($p < 0.001$; $p < 0.05$), with no differences between samples from the control and MNP with zinc groups. Similarly, the carriages of three mucormycete genera (*Rhizomucor*, *Actinomucor*, and *Mucor*) were 1.3, 1.5, and 1.8-fold higher, respectively, in the MNP group compared to the control ($p < 0.01$; $p < 0.05$; $p < 0.05$), with no significant differences between the control and MNP with zinc groups. *Toxoplasma* was the only genus with significantly reduced carriage in children receiving MNPs with zinc, with 11 and 23-fold reductions compared to CTL and MNP groups ($p < 0.05$, $p < 0.001$); however, we observed non-significant reductions in other organisms such as *Cercomonas* and *Mucor* (2 and 1.4-fold, respectively) suggesting possible species-specific effects. Despite previous reports of the impact of zinc on helminths[24], we did not detect significant differences in the carriage of the tapeworm *Hymenolepis nana* among treatment arms.

**Micronutrient supplements are associated with specific bacterial communities.** In addition to profiling eukaryotic communities, we were interested in understanding whether micronutrient supplements affected the bacterial microbiome. We applied 16S rRNA amplicon sequencing, to profile the stool bacterial microbiota of all 160 samples. From the 13,984,120 sequenced reads (median 92,628), we identified 1108 bacterial OTUs across all 160 samples (median 50; Supplementary Data 4, Supplementary Data 5). Actinobacteria and Firmicutes were found to dominate with just two OTUs (both assigned to *Bifidobacterium*) accounting for over 50% of all reads (Fig. 3a). Age was the primary determinant of bacterial richness (GLM, $\beta = 43.65$, 95% CI [31.98, 55.31], $p < 0.001$) and evenness (GLM, $\beta = 0.80$, 95% CI [0.59, 1.02], $p < 0.001$) (Fig. 3b, Supplementary Fig. 4), as well as patterns of taxonomic composition as measured by Bray–Curtis and weighted UniFrac dissimilarities (adonis, $R^2 = 0.06$, $p < 0.001$; $R^2 = 0.05$, $p < 0.001$; Fig. 3c). Regression of compositional differences in each child at 12 and 24 months using

partial correspondence analysis indicated that 56% of Bray–Curtis and 59% of weighted UniFrac changes may be attributed to age. By correlating the abundances of bacterial taxa with the first two axes of the Bray–Curtis ordination, we identified the candidate drivers of community differences as the two dominant *Bifidobacterium* species, with opposite abundance patterns perhaps suggesting succession of one species by the other.

Consistent with a previous study[28], bacterial richness was reduced in undernourished children (GLM, $\beta = -29.19$, 95% CI [−52.99, −5.39], $p < 0.05$), while a significant interaction between nutritional status and place of residence indicated that bacterial evenness was reduced in undernourished children from the urban setting (GLM, $\beta = 1.03$, 95% CI [0.11, 1.95], $p < 0.05$) (Fig. 3b, Supplementary Fig. 4b). We detected no significant association between nutritional status or locality and taxonomic composition in this cohort.

Interestingly, gut bacteria were also influenced by micronutrient supplements. Treatment with MNPs was associated with an overall increased abundance of Actinobacteria in children at 12 months compared to the control group and those receiving MNPs with zinc (GLM, $\beta = 36020$, 95% CI [7239, 64802], $p < 0.05$), but reduced abundance in the MNP group at 24 months (GLM, $\beta = -52670$, 95% CI [−93373, −11966], $p < 0.05$) (Fig. 3d). Firmicutes were reduced in the presence of zinc in both age groups (GLM, $\beta = -261976$, 95% CI [−476591, −47362], $p < 0.05$), with a non-significant reduction in those supplemented without zinc (GLM, $\beta = -206413$, 95% CI [−416049, 3221], $p = 0.055$). Supplementation tended to reduce overall bacterial richness with an effect that reached significance in the MNP group (GLM, $\beta = -14.66$, 95% CI [−29.01, −0.31], $p < 0.05$) (Fig. 3b) and influenced taxonomic composition as measured by weighted UniFrac (adonis, $p < 0.01$, $R^2 = 0.03$) but not Bray–Curtis dissimilarities. Specifically, phylogenetic variance, defined as dispersions from the group centroids, differed among groups (two-way ANOVA, $p < 0.001$), with significantly smaller differences among 12-month-old children receiving MNPs and MNPs with zinc (Tukey HSD post hoc, $p < 0.01$) (Fig. 3e, Supplementary Fig. 4c). This may suggest that micronutrients support or restrict the growth of select taxa. Through differential abundance analysis, we identified 14 taxa with significantly reduced abundances in both supplemented groups at 12 months compared to controls, including over 10-fold reductions in *Anaerostipes*, *Anaerosalibacter*, and *Clostridium XI* (DESeq2, Wald test, $p < 0.05$; Fig. 3f). Two additional *Anaerostipes* OTUs were significantly reduced in supplemented groups at both ages (DESeq2, Wald test, $p < 0.05$), with six OTUs reduced at 24 months only. MNPs with zinc were associated with significant changes in an additional 46 taxa, and 29 taxa were altered in MNP samples (DESeq2, Wald test, $p < 0.05$). These included a seven-fold increase in *Escherichia-Shigella* abundance in 12-month-old MNP-supplemented children, increases in several Lactobacilli, and a 1.3-fold reduction in one *Bifidobacterium* OTU (Fig. 3g). These data reveal that micronutrient supplementation may impact bacterial communities during early development.

**MNPs may destabilize microbial interactions in undernourished infants.** In the previous analyses, we were concerned with global changes in community structure, as well as individual taxa. To explore interactions between taxa and how these may change in response to developmental, nutritional, or environmental perturbations (i.e., age, nutritional status, supplementation, and locality), we constructed microbial interaction networks (Fig. 4). Interactions between taxa were inferred from pairwise associations (i.e., co-occurrence or mutual exclusion), based on their abundance in children grouped by one or more explanatory

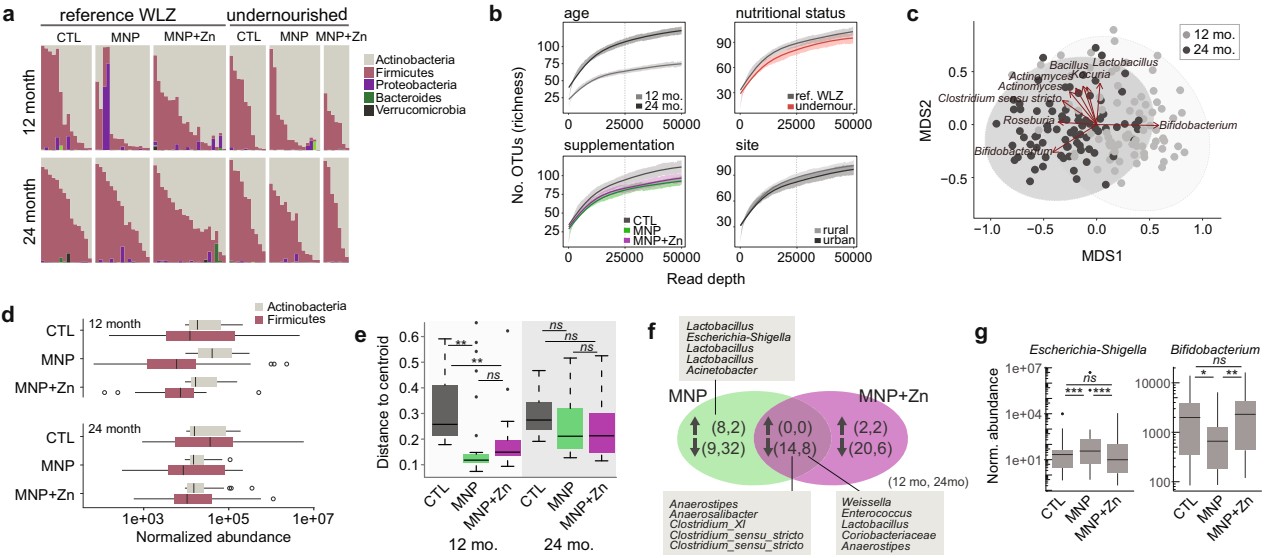

**Fig. 3 Bacterial microbiota change with age and supplementation. a** Relative abundances of bacterial phyla in 12 (top) and 24 (bottom) month old children based on 16S data. **b** Rarefaction curves comparing mean species richness by age, micronutrient supplementation, nutritional status, and place of residence (site). Shaded regions represent standard errors and the dotted lines denote the read depth at which significance was tested. **c** Non-metric multidimensional scaling of bacterial compositions based on Bray–Curtis dissimilarities. Samples are colored by age, and ellipses represent 95% confidence intervals. Arrows indicate the direction of increasing abundance of the top 10 bacterial OTUs significantly correlated with the ordination axes, scaled by their strength of correlation ($r^2$). **d** Mean DESeq2-transformed abundance of Actinobacteria and Firmicutes in samples ($n = 80$) grouped by age and treatment. **e** Compositional variance among samples grouped by supplementation arm and age, based on weighted UniFrac dissimilarities and measured as distances to the centroid. Significance was tested using two-way ANOVA with the Tukey HSD post-hoc test ($n = 80$, 12 months CTL vs MNP $p = 0.0032$, CTL vs MNP + Zn $p = 0.0029$) **f** Venn diagram shows numbers of bacterial taxa significantly increased or decreased in abundance, as indicated by arrows, in supplemented groups relative to the control. Numbers in brackets refer to taxa at 12 and 24 months of age. Select taxa are listed. **g** Normalized abundance of *Escherichia–Shigella* and *Bifidobacterium* OTUs across supplementation arms at 12 months ($n = 80$). Significant differences were determined using the Wald test in DESeq2 and Benjamini–Hochberg multiple testing correction (*Escherichia–Shigella* CTL vs MNP $p = 2.6e-11$, MNP vs MNP + Zn $p = 7.1e-17$; *Bifidobacterium* CTL vs MNP $p = 0.029$, MNP vs MNP + Zn $p = 0.0064$). Boxplots in **d**, **e**, and **g** represent medians and interquartile ranges (IQR), and whiskers demarcate 1.5x IQR. Asterisks in (**e** and **g**) are *$p < 0.05$, **$p < 0.01$, ***$p < 0.001$, ns not significant.

variables. Based on these associations, graphical networks were built depicting a community of eukaryotes and bacteria as nodes, with associated taxa (those thought to influence one another's growth) linked by edges. We compared the network topologies to evaluate putative differences in microbial communities. We found that the numbers of interactions held by taxa (node degree) were significantly higher in the older age group irrespective of treatment, nutritional status, or place of residence (Fig. 4a), consistent with the development of more complex communities as the child matures[29]. The greatest difference, with a 2.5-fold increase ($p < 0.001$), was noted in the networks of children in the MNP arm, which had the fewest taxon interactions at 12 months (MNP vs. CTL, $p < 0.001$; MNP vs. MNP with zinc, $p < 0.01$), but achieved parity with the control and MNP with zinc groups by 24 months (MNP vs. CTL, *ns*; MNP vs. MNP with zinc, *ns*), 6 months after supplementation had ceased. Cross-kingdom interactions between bacteria and eukaryotes represented 20 to 30% of all network interactions at 12 months, falling to between 15 and 24% by 24 months of age, a decrease that was significant in undernourished children (Fisher's Exact, OR 2.1, 95% CI [1.3, 3.3], $p < 0.05$; Fig. 4b).

The impact of supplementation on the microbial interactions appeared to further depend on the health of the host. When split by nutritional status, we observed important differences in the microbial networks of 12-month-old undernourished infants supplemented with micronutrients compared to the control and reference WLZ groups (Fig. 4c–e). Within control groups, the microbial networks of undernourished infants and those within the reference WLZ group had similar levels of connectivity, with non-significant differences in degree distribution (Fig. 4c) and

betweenness centrality scores (Fig. 4d). While children in the reference WLZ group receiving either supplement were associated with small but significant reductions in microbiota betweenness (MNP, $p < 0.05$; MNP with zinc, $p < 0.01$), greater reductions were observed in supplemented undernourished children ($p < 0.001$). Since betweenness provides a measure of the degree of coordination within a network, these findings suggest that micronutrient supplementation, with or without zinc, may result in microbial communities that are less organized at 12 months of age. This is further illustrated by the network visualizations (Fig. 4e), where, in addition to changes in network density, we also identified shifts in taxa with the highest betweenness values (which can be interpreted as those taxa most likely to mediate important coordinating roles within the communities). For example, within the control group, Clostridia, two species of Mucoromycota and the ciliate *Bromeliothrix* occupy central roles in the network of reference WLZ infants, while in undernourished infants, these central roles are held by *Trichosporon*, *Longamoeba*, and *Prevotella*. In supplemented reference WLZ groups, Bacilli exhibit the highest betweenness values in the absence of zinc, while these are replaced by Proteobacteria in the communities from infants receiving MNPs with zinc. However, within undernourished infants receiving either supplement, microbial networks appear largely fragmented (Fig. 4e), with dramatically lower degree distributions and betweenness compared to the control group, (Fig. 4d) suggesting that micronutrient powders may have a disruptive effect on a fragile microbial community. Comparison of microbial networks by location of residence further showed an increased density of interactions within each rural group (control or supplemented) compared to

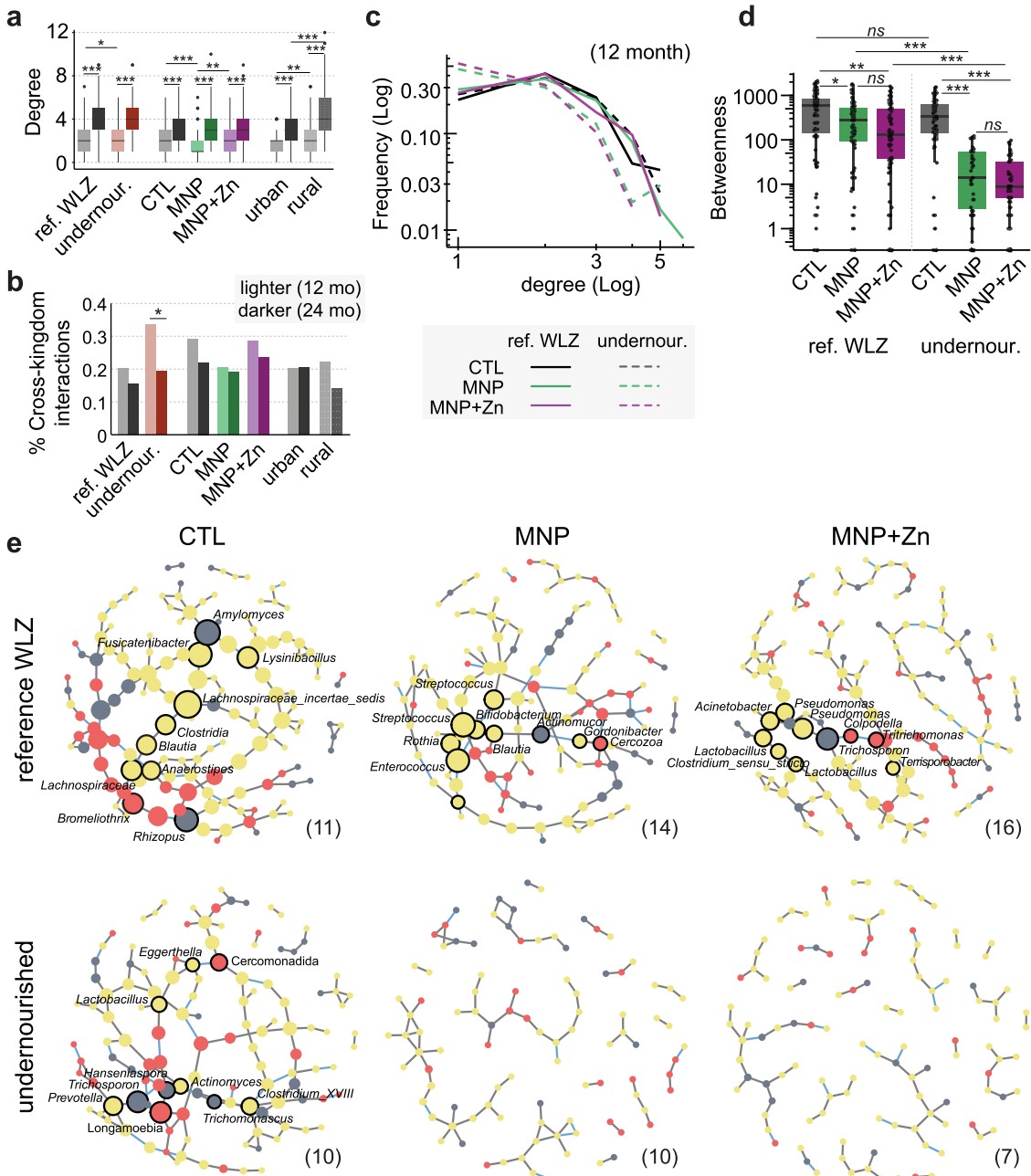

**Fig. 4 Supplementation influences microbial interactions. a** Density of microbial interactions at 12 and 24 months of age, calculated as the number of significant associations (node degrees) each microbe has with other taxa within a sample group. Samples were grouped by age and nutritional status, supplementation arm, or place of residence (site). Lighter and darker hues represent data from 12 ($n = 68$) and 24-month-old children ($n = 71$), respectively. *P*-values are provided in Supplementary Data 3. **b** Proportions of significant microbial interactions occurring cross-kingdom at 12 and 24 months of age, within indicated sample groups (12 months $n = 68$; 24 months $n = 71$; undernourished 12 mo. vs. 24 mo. $p = 0.021$). **c** Degree distribution and **d** betweenness centrality scores of microbial networks in 12-month-old children grouped by nutritional status and supplementation arm ($n = 68$; nutritional status (ref.WLZ vs undernourished) in MNP $p = 9.8e-9$ or MNP + Zn $p = 2.2e-9$; supplementation (CTL vs MNP) in ref.WLZ $p = 0.033$ or undernourished $p = 5.6e-10$, and (CTL vs MNP + Zn) in ref WLZ $p = 0.0027$ or undernourished $p = 4.2e-11$). **e** Graphic representations of networks representing significant microbial interactions in 12-month-old children, grouped by nutritional status and micronutrient treatment. Nodes represent bacterial OTUs (yellow) and protozoan and fungal genera (red and gray, respectively), scaled by betweenness centrality scores; node borders denote taxa with the ten highest scores. Edges represent significant positive (gray) and negative (blue) correlations among microbiota. Taxa with no predicted interactions have been removed. The numbers of samples used to generate each network are indicated within brackets. Boxplots in **a** and **d** represent medians and interquartile ranges (IQR), and whiskers demarcate 1.5 × IQR. Statistical differences were determined using a two-sided Wilcoxon rank-sum test in (**a** and **d**) or a two-sided Fisher's Exact test in (**b**) and corrected for multiple testing using the Benjamini–Hochberg procedure. Significance \*$p < 0.05$, \*\*$p < 0.01$, \*\*\*$p < 0.001$, ns not significant.

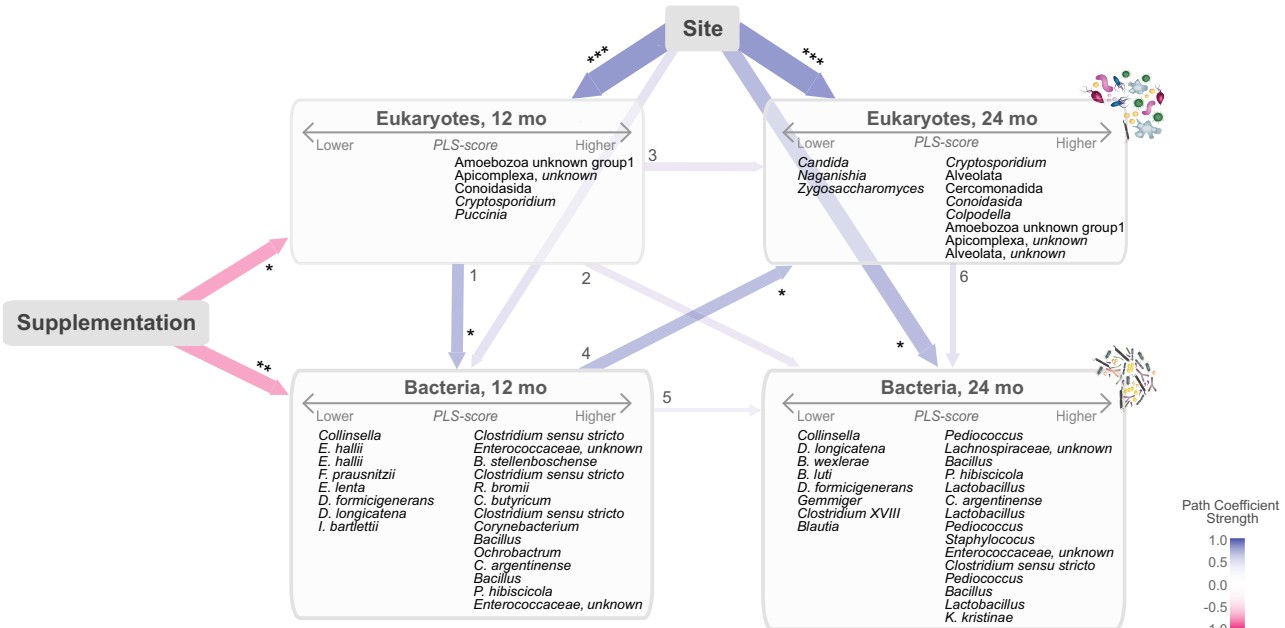

**Fig. 5 Graphic representation of the cross-associations among demographic variables, micronutrient supplementation, and microbiota at 12 and 24 months.** Interconnected arrows indicate the tested cross-correlated paths between: place of residence (site), supplementation, and the community patterns of bacterial and eukaryotic OTUs summarized as latent PLS scores. Paths represent composites of direct and indirect effects among variables. Negative correlations are indicated in pink and positive in blue. Arrow thickness is weighted by the coefficient strength of the direct effect as detailed in Supplementary Data 6. Significance of direct effects, $*p < 0.05$, $**p < 0.01$, and $***p < 0.0001$. OTUs that loaded positively (>0.4) or negatively (<−0.4) within each PLS-score are listed within boxes. PLS, partial least square; OTUs, operational taxonomic units.

the urban groups (Supplementary Fig. 5), however, our ability to detect microbial interactions in the urban cohort may have been hampered by a lower representation of participants from this district. Low participant numbers precluded us from successfully generating networks at 24 months, where numbers of microbial taxa are greater. Data from additional timepoints are needed to confirm the potentially destabilizing effects of MNPs throughout the intervention and the putative recovery after the intervention has ceased.

**Micronutrient supplementation and place of residence impact cross-kingdom microbial interrelationships in the early infant gut.** Based on our findings described above, we hypothesized that the effects of supplementation on microbial communities at 12 months could also indirectly influence later microbial profiles. We further hypothesized that early exposure to eukaryotes (before or at 12 months of age) could alter age-related differences in the bacterial microbiome. To explore these complex interrelationships, we generated an integrated model of cross-correlations among microbial profiles and explanatory variables using partial least squares (PLS) path modeling (Fig. 5, Supplementary Data 6). The identified relationships, indicated by arrows, are composites of direct and indirect effects, where indirect effects estimate the impact of an upstream factor (e.g., supplementation) on a downstream element (e.g., eukaryotes at 24 months) via an intermediate (e.g., bacteria at 12 months). We found that micronutrient supplementation influenced components of eukaryotic and bacterial composition at 12 months (path coefficient $-0.27 \pm 0.11$, $p < 0.05$; path coefficient $-0.27 \pm 0.09$, $p < 0.01$), with possible indirect carryover effects to microbial composition at 24 months (i.e., supplementation may influence microbial profiles at 24 months via its early impact at 12 months; indirect effects of $-0.11$, Supplementary Data 6). Consistent with our findings above, abundances of *Mucor* and *Euglyphida*

correlated with supplementation at 12 months (cross-correlation coefficients $-0.35$ and $-0.34$, respectively; Supplementary Data 6). Eukaryotic profiles at 12 months of age were associated with changes in abundance of certain bacteria at 12 months (Fig. 5, arrow 1; path coefficient $0.27 \pm 0.12$, $p < 0.05$) and, these changes, in turn correlated with shifts in eukaryotic composition at 24 months (Fig. 5, arrow 4; path coefficient $0.21 \pm 0.095$, $p < 0.05$). Place of residence also significantly influenced microbial profiles, with the greatest effects observed on eukaryotic profiles (12 months, path coefficient $0.52 \pm 0.09$, $p < 0.0001$; 24 months, path coefficient $0.48 \pm 0.1$, $p < 0.0001$). Children from the rural community had increased levels of several alveolates, including *Cryptosporidium*, at both ages (cross-correlation coefficients: 12 months, 0.40; 24 months, 0.69). These children also had higher abundances of several *Clostridium* OTUs at both ages and sustained higher levels of *Lactobacillus* at 24 months. A stratified analysis of reference WLZ and undernourished children did not reveal groups to have significant differences in their correlation structure with variables (i.e., path coefficients were not significantly different between groups). However, the stability of path coefficients could not be confirmed in our cohort using bootstrapping, a robust second-step validation procedure that iteratively recalculates path coefficients from a random sample of participants. With our sample size, bootstrapping produced large standard errors likely resulting from unevaluated interactions and/or diverse microorganism carriage (Supplementary Data 6). This may also suggest that the stochastic presence/absence of certain eukaryotes could be important in driving the supplement-related changes in the microbiome. Nevertheless, given the consistency with our earlier findings, this model supports the need to further investigate the influence of MNP supplementation on the relationships of eukaryotic and bacterial communities in the developing gut of vulnerable infants who carry diverse eukaryotic organisms.

## Discussion

Malnutrition, both undernutrition and obesity, are associated with altered bacterial compositions, wherein the former, under-developed bacterial communities have the capacity to induce weight loss[6,30]. Here, we have shown that the gut microbiota of both undernourished children and those within a healthy weight range includes a diverse group of protozoa, helminths, and fungi, each with the capacity to impact host health. We have also shown that supplementation with MNPs, a strategy used to improve growth and alleviate micronutrient deficiencies[14,16], may influence the development of the microbiome in these susceptible populations.

Consistent with previous studies, we found that bacterial communities became more complex with age. Eukaryotic communities, however, were not significantly impacted by age but instead were associated with micronutrient supplementation and place of residence. Only the tapeworm *H. nana* was identified at significantly higher levels in undernourished children. While *H. nana* infection is usually asymptomatic, high egg burdens in children have previously been associated with diarrhea, abdominal pain, and weight loss[31], with exacerbated morbidity in children <5 years[32]. We associated rural habitation with significantly more diverse protozoan communities and, in particular, increased prevalence of *Cryptosporidium*. An important cause of infant mortality and childhood malnutrition, *Cryptosporidium* infection is attributed to unsafe drinking water and inadequate sanitation often associated with rural settings[27,33]. While approximately half of all children enrolled in the parent trial had access to piped drinking water (41 and 52% in the urban Bilal colony and rural Matiari sites, respectively), only 4% of children in the Matiari district had access to underground sewage, compared to 95% in the Bilal Colony[27], consistent with a lack of wastewater sanitation resulting in higher parasite carriage. While the Global Enteric Multicenter Study (GEMS), the largest childhood diarrhea study to date, with 22,568 children enrolled across seven developing countries in Asia and Africa, reported *Cryptosporidium* as a leading cause of death in 12–23-month-old children with moderate to severe diarrhea[34], we found a high prevalence of this parasite in the absence of diarrhea (80 and 83% at 12 and 24 months in the Matiari district, and 60 and 33% in the Bilal urban colony). As our detection is based on 18S rRNA amplicon sequencing, we may have detected a broader group of species with variable pathogenic potential compared to the GEMS study, which applied a specific oocyst antigen immunoassay. Alternatively, our findings may indicate a high prevalence of asymptomatic infections, with symptomatic infections resulting from additional unknown factors[7,35]. The prevalence of *Cryptosporidium* in our cohort was also higher than previously reported in non-diarrheal stools, using oocyst antigen testing, in the neighboring Naushero Feroze District (5.1% between 12 and 21 months of age), where *Cryptosporidium* contributed to 8.8 diarrheal episodes per 100 child-years[36,37]. This study associated asymptomatic enteropathogen infection, including *Cryptosporidium* and *Giardia*, across eight countries with elevated inflammation and intestinal permeability, factors thought to increase risk of stunting and impact the effectiveness of nutritional interventions in low-resource settings[37].

A major focus of our study was to estimate the effect of micronutrient supplementation on the gut microbiota. We found that children receiving supplements without zinc were associated with distinct eukaryotic communities, featuring an increased prevalence of multiple protozoa and fungal taxa, including species with known potential to cause symptomatic infections and disrupt the gut microbiome; however, the addition of zinc to these supplements alleviated these increases, while significantly reducing the prevalence of *Toxoplasma* and overall protozoan richness. These findings are consistent with a previous report which suggested that zinc has a parasite-specific protective effect against infection and ensuing diarrhea[24]. Fungal diversity was not impacted by age, supplementation, place of residence, or nutritional status. However, the predominance of Mucoromycota, particularly in children receiving MNPs without zinc, is of concern, as these organisms are responsible for rare but lethal invasive fungal infections that have previously been reported in low birth weight infants and malnourished children[38]. Although incidence of infections is rising globally, rates of mucormycoses are particularly high in Asia[39]. Notably, a recent spike in infections, also termed Black fungus, in thousands of active and recovered Covid-19 patients in India was attributed to treatment with corticosteroids to control inflammation, in conjunction with a high prevalence of diabetes[40].

It has been well established that iron supplementation can promote the virulence of particular fungi and parasites[41,42]. Several studies have shown that iron alone or in combination with other micronutrients worsens existing infections, lengthens the duration and severity of diarrhea and increases mortality rates in children[22,27,41]. Consequently, sequestration of free iron by host proteins such as lactoferrin is a key defense mechanism to limit growth of pathogens, including Mucorales[43]. Iron deficiency has furthermore been suggested as protective against malaria infection[44,45], and provision of supplements containing iron in endemic regions has been cautioned against due to increased malaria-related hospitalization and mortality of children[41]. While a deficiency in zinc has been associated with impaired immune function and susceptibility to enteroinfections[46], supplementation in the context of enteric pathogens was shown to have parasite-specific outcomes. Provision of zinc alone can increase the incidence of *Ascaris lumbricoides* and duration of *Entamoeba histolytica* infections, but it has also been shown to reduce the duration of associated diarrheal episodes as well as lower the prevalence of *Giardia lamblia* infections[24]. Interestingly, asymptomatic *Giardia* infections in children in Tanzania were associated with reduced rates of diarrhea and fever, an effect which was lost in children receiving vitamin and mineral supplements, including both iron and zinc[47]. Our data suggest that while iron, vitamins, or both, may promote growth and survival of commensal and potentially pathogenic eukaryotes, resulting in a shift in eukaryotic community structure and disrupting the gut environment, the addition of zinc may reduce the ability of at least some eukaryotic microbes to infect and persist. The findings of reduced bacterial diversity in 12-month-old infants receiving micronutrient supplements, together with elevated levels of *Escherichia–Shigella* and reduced beneficial *Bifidobacterium*, are also consistent with previous reports, where reductions in beneficial bifidobacteria and lactobacilli and increased enterobacteria in infants receiving iron-containing micronutrients were linked to an elevated risk of inflammation and diarrhea[22,23,48]. The parent cRCT associated *Aeromonas* infection with increased diarrhea in MNP-supplemented groups[27]. We did not detect this bacterium in our data, possibly due to the exclusion of diarrheal samples.

The impact of micronutrient supplementation also extended to the structure of the microbial communities. Microbial networks, representing significant correlations in the co-occurrence or mutual exclusion of specific bacteria and eukaryotes, revealed higher network connectivity in the control groups, with the networks generated from the undernourished infants receiving both types of supplements revealing a more fragmented structure. This fragmentation suggests a destabilization of species interactions within the developing gut microbiota in undernourished infants. Possibly contributing to this disruption of community associations is the presence of specific eukaryotic microbes, as evidenced by the increased carriage of a number of protozoan and

fungal taxa in children receiving MNPs, and/or the expansion of pathogenic bacteria. These microbes may interfere with the maturation of commensal bacteria through predation, competition for resources, and/or modulation of host immunity. In undernourished infants, the cumulative effect of increases in pathogenic organisms on community structure may be more pronounced than in infants within a healthy weight. Enteropathogens *Giardia lamblia* and enteroaggregative *Escherichia coli*, for example, were shown to have a greater impact on growth in protein-deficient mice during co-infection, an effect which was dependent on the resident gut bacteria[49]. Taken together, our data showing increased carriage of eukaryotic microbes and increased abundance of *Escherichia–Shigella* in children supplemented with micronutrients, as well as a potential loss of organization in microbial interactions in supplemented undernourished children, may offer at least a partial explanation for previous reports of increased duration and severity of diarrhea as well as increased intestinal inflammation in children supplemented with micronutrient powders[27].

Due to the relatively small numbers of samples, we were unable to generate separate networks for the three treatment arms for 24-month-old children, or further segregate the networks by place of residence. We note that supplementation ceased at 18 months of age, consequently the acute effects of these supplements may have dissipated prior to 24 months. Moreover, micronutrient interventions may impact undernourished children differently in settings associated with high *Cryptosporidium* burden, for example. Networks constructed from small sample sizes also have a greater propensity for prediction of spurious interactions and a reduced capability to detect true associations[50]. In absence of sample limitations, network reconstructions from larger balanced groups, with additional timepoints, could improve our ability to discern key microbial linkages impacted by interventions during different stages of growth and weaning. We note that lack of spike-in controls in this study did not allow us to estimate the absolute abundances of microbiota. In future studies, such an approach could confirm whether supplementation, for example, influences the relative representation of eukaryotes and bacterial cells in the gut. The notable absence of *Giardia*, a parasite typically prevalent in this demographic, is likely due to mismatches to the 18S rRNA sequencing primers[13]. Parasite diagnostic data from the trial did identify *Giardia* in 37 infants at 12 months and *Cryptosporidium* in seven, but noted no significant increases in either of the supplemented groups[27]. Prevalence was nearly two-fold higher at the rural site, consistent with our findings for *Cryptosporidium*, emphasizing the need for location-specific investigations of the effects of micronutrient supplements. In addition to potential intraspecies variation, our detection of high sequence diversity in *Cryptosporidium* OTUs specifically, and eukaryotic taxa in general, may be exaggerated by a high proportion of non-overlapping amplicon reads, a consequence we have attempted to minimize through manual curation. Regardless, we report that eukaryotic microbiota are abundant members of the gut microbiome even in infancy, and given the known role of parasitic pathogens in diarrheal disease and the association of fungi with obesity and inflammatory bowel disease[51,52], their role in malnutrition should be further studied.

Our integrated model of microbial relationships and influencing external factors recapitulated a number of key earlier findings, including the impact of micronutrients and locality on gut eukaryotes. Furthermore, it supports the notion that complex cross-kingdom interactions may influence the gut bacterial composition and provides a valuable framework to dissect the direct and indirect effects of eukaryotic infections or nutritional interventions on the maturing gut microbiome. The instability observed in the pathway coefficients of the model, when participants were randomly resampled, is likely the result of the heterogeneous and stochastic nature of eukaryotic carriage (i.e., random exclusion of participants carrying eukaryotes with the strongest responses to supplementation or locality) and ultimately our small sample size. However, given the current debate concerning the use of MNP supplementation and its role in gastrointestinal disorders[53,54], such a framework is expected to play a key role in identifying scenarios where MNP supplementation may require more cautious thinking.

This study demonstrates that micronutrient powders impact the infant microbiota, with potentially disruptive effects driven through the promotion of specific organisms during the early stages of microbiome development. These findings are of relevance to micronutrient supplementation strategies, especially those targeting vulnerable children in low-resource settings.

## Methods

**Study design and participant selection**. All stool samples and clinical data analyzed in this study were collected in a cluster randomized controlled trial (cRCT) (ClinicalTrials.gov identifier NCT00705445 [https://clinicaltrials.gov/ct2/show/NCT00705445]) of micronutrient powders (MNPs) with or without zinc that was conducted in urban (Bilal colony, squatter settlement within Karachi) and rural (Matiari district, ~200 km from Karachi) communities in Sindh, Pakistan between November 2008 and December 2011[27]. In brief, a total of 2746 children (control ($n = 947$), MNP without zinc ($n = 910$), and MNP with zinc ($n = 889$)) were enrolled in the parent cRCT at 6 months of age (Supplementary Table 1). Participants in the control group received health promotion messaging surrounding recommended feeding practices, whereas participants randomly assigned to MNP groups received the same messaging and were given a 14-day supply of MNPs (replenished every 2 weeks between 6 and 18 months of age). Families were instructed to mix one sachet of MNPs per day into the child's regular foods. MNPs contained vitamins A, C, D, folic acid, and microencapsulated iron, with or without 10 mg zinc. All children were prospectively followed until 24 months of age, and clinical data were collected every 2 weeks starting at 6 months of age with anthropometric assessments conducted monthly, as previously described[27]. Routine stool samples were collected from participants at enrollment, 6, 12, 18, and 24 months of age. Caregivers were provided with a stool collection kit (plastic diaper, collection container, spatula, and gloves) and training on how to self-collect stool, document the time of defecation, store the sample in a shaded area of the household and notify study staff. Stool samples were picked up at participants' homes within 1 h of defecation and immediately transported to the field research office in a 2–4 °C cold box. Stool was divided into up to eight aliquots (~800 mg to 1 g per aliquot), of which 5–6 were marked for future analyses. Aliquots were then transported to Aga Khan University (AKU) within 4–6 h and frozen without additives at −80 °C. Samples analyzed in this study were shipped from AKU to the Hospital for Sick Children (SickKids) on dry ice, where they were stored at −80 °C until analysis.

Eighty children were selected from all three supplementation arms of the parent cRCT for microbiome profiling (Supplementary Fig. 1). The selection criteria were: (1) having stool samples collected at 12 and 24 months of age available and archived at −80 °C; (2) having no record of antibiotic administration within 14 days of stool sample collection; (3) having at 24 months a weight-for-length z-score (WLZ) < −2 below the median (undernourished) or > −1 (reference WLZ) based on WHO 2006 growth references (www.who.int/childgrowth); and, (4) no reported diarrhea within 7 days of stool collection. Participants within the reference group were further selected to have the fewest WLZ scores < −1 at other time points to represent as healthy as possible a comparator group. Four participants (three from the reference group and one undernourished) with no detectable 18S amplicons were excluded due to lack of data. A further two participants from the reference group were excluded to meet ethics criteria restricting the analysis to the sequencing of 160 samples. Participant characteristics for the microbiome substudy were summarized as medians with interquartile ranges (IQRs) or means with standard deviations (SD) if continuous variables, and percentages if categorical in Table 1 and Supplementary Table 1. For comparison, the characteristics of all participants followed to 24 months of age in the parent cRCT are summarized in Supplementary Table 1. A CONSORT checklist is provided in the supplementary material.

**DNA extraction and amplicon sequencing**. DNA was extracted from 100–200 mg of stool using the E.Z.N.A.TM Stool kit (Omega Bio-Tek Inc, GA, USA) according to the manufacturer's protocol. Mechanical disruption of cells was carried out with the MP Bio FastPrep-24 for five cycles of 1 min at 5.5 M/s. 16S variable region 4 (V4) amplifications were carried out using the KAPA2G Robust HotStart ReadyMix (KAPA Biosystems) and barcoded primers 515 F (GTGCCAGCMGCCGCGGTAA) and 806 R (GGACTACHVGGGTWTCTAAT)[55]. The cycling conditions were 95 °C for 3 min, 22 cycles of 95 °C for 15 s, 50 °C for 15 s and 72 °C for 15 s,

followed by a 5 min 72 °C extension. Libraries were purified using Ampure XP beads and sequenced using MiSeq V2 (150 bp × 2) chemistry (Illumina, San Diego, CA). 18S V4 + V5 amplification was achieved using the iProof DNA polymerase (Bio-Rad Laboratories, Hercules, CA) with primers V4-1 (GCGGTAATTCCAGC TC) and V4-4 (GCCMTTCCGTCAATTCC) as previously described[13]. Briefly, the cycling conditions used were 94 °C for 3 min, 30 cycles of 94 °C for 45 s, 56 °C for 1 min, and 72 °C for 1 min, followed by a 10 min 72 °C extension. Barcodes were ligated, and libraries were sequenced using MiSeq V3 (300 bp × 2) chemistry (Illumina, San Diego, CA). Sequencing was performed at the Centre for the Analysis of Genome Evolution and Function (Toronto, Canada).

**Sequence data analysis**. 16S data were quality filtered and processed within a cluster computing environment based on Lenovo SD530 Intel-based servers running Linux (CentOS 7), using VSEARCH v2.10.4[56] and the UNOISE pipeline in USEARCH v11.0.667[57,58]. Filtered sequences were clustered to 99% sequence identity, and the resulting operational taxonomic units (OTUs) were classified with a minimum confidence of 0.8 using the SINTAX[59] algorithm and the Ribosomal Database Project version 16[60].

18S data were quality filtered using Trimmomatic v0.36[61] and read pairs with a minimum 200 nucleotide length were merged using VSEARCH or artificially joined using a linker of 50 ambiguous nucleotides ($N_{50}$) using USEARCH. Resultant amplicon sequences were clustered to 97% sequence identity using the UCLUST[57] algorithm and taxonomically classified using SINA v1.2.11[62] with a minimum 90% sequence similarity threshold. Unclassified sequences were submitted for classification using SINTAX and the SILVA v132 non-redundant reference database[63], and those still unclassified were compared to the NCBI non-redundant nucleotide database[64] (downloaded Nov 28, 2017) by BLAST 2.2.28[65] using a 90% cutoff for both sequence identity and query coverage. Phylogenetic trees were constructed from 16S and 18S OTUs using FastTree 2.1[66], and visualized using the Iroki viewer[67]. Prevalences were shown for taxa with a minimum detection of five reads.

**Microbial diversity and differential abundance analyses**. Microbiota richness (number of OTUs) and evenness (Shannon Diversity Index, H) were calculated using Phyloseq 1.20.0[68]. Rarefaction curves for each were generated at 100 read intervals to a maximum of 5000 or 50,000 for eukaryotes and bacteria, respectively, and sample group means and standard errors were plotted. As the intra-individual correlation of richness and evenness at 12 and 24 months was low, we implemented generalized linear models (GLMs) using the glm function in R to test for group differences, using diversity values averaged from 100 independent rarefactions at read depths of 25,000 (bacteria) or 1000 (protozoa and fungi). Read depths were chosen to ensure sufficient sampling of OTUs while maintaining a representation of samples, a concern for eukaryote data where read depths were highly variable (Supplementary Fig. 6). A read depth of 1000 for eukaryotes resulted in a loss of 11 (of 160) samples, while a read depth of 25,000 for prokaryotes resulted in a loss of ten samples. We performed stepwise model selection using AIC with MASS[69] to identify which of the following explanatory variables best explain diversity: age, nutritional status (defined as undernourished, WLZ < −2, or reference, WLZ > −1), supplementation, and urban versus rural site.

Differences in bacterial composition (beta diversity), based on Bray–Curtis and weighted UniFrac dissimilarity scores, were calculated with Phyloseq and vegan 2.5–7[70]. OTU counts were normalized using the median of ratios method in DESeq2, and only taxa represented by a minimum of five reads in at least 5% of the samples were used in the analysis. The contribution of age to bacterial composition was calculated using the capscale function, and the remaining variables were tested for significance in age-stratified samples using adonis. The compositional variance within groups, measured as distances to centroids, was evaluated using the betadisper function and tested for significance using two-way ANOVA, with pairwise differences delineated using the Tukey HSD post-hoc test. All adonis and betadisper tests were carried out with 9999 permutations. We applied non-metric dimensional scaling (NMDS) to ordinate samples based on their compositional dissimilarity. The envfit function was used to identify taxa whose abundance significantly correlated with the first two ordination axes (candidate drivers of community differences), projected as vectors in the direction of change in abundance and scaled by the root square of the correlation. Differences in protozoan and fungal compositions were evaluated at 1000 read depth based on unweighted UniFrac distance scores, and significance was tested as above. Sample distances were visualized using NMDS (calculated over five dimensions, with 50 random starts) with candidate drivers of community differences identified using the envfit function and projected as vectors, as for bacterial data. Differential taxon abundance was tested with DESeq2 1.22.2[71] in samples containing a minimum of 1000 reads.

Two-sided Fisher's Exact or pairwise test from the rstatix 0.7.0 package was used to evaluate differences in eukaryote carriage among participant groups, using a minimum five read detection threshold per OTU and grouping OTUs to the genus level or the lowest assigned taxonomic level. Benjamini–Hochberg correction was applied for multiple testing[72].

**Microbial interaction networks**. Microbial interactions were inferred from pairwise associations between taxa in children, grouped by one or more explanatory variables (age, nutritional status, supplementation, and place of residence). Briefly, bacterial and eukaryotic datasets were first rarefied to 25,000 and 1000 reads, respectively. As before, thresholds were selected to balance sufficient sampling of OTUs with maximizing sample group sizes (Supplementary Fig. 6); eukaryotes were agglomerated to the genus or lowest assigned taxonomic level. Pairwise associations between taxon abundances were calculated for eukaryotic and bacterial datasets simultaneously using SpiecEasi 1.1.0[50], with the neighbor selection (MB) method, nlambda 100 and lambda.min.ratio 1e-02 parameters, and used to construct graphical networks using igraph 1.2.6[73]. Interaction networks were compared among groups of children using topological properties, including degree centrality (the number of significant associations held by each taxon in a network) and betweenness centrality (the number of paths linking each pair of microbes in a network that pass through a particular taxon). Betweenness centrality can be interpreted as the importance of a particular taxon in mediating overall microbial interactions. Differences in degree and betweenness centralities were tested using the Wilcoxon rank-sum test, and proportions of cross-kingdom interactions among groups were evaluated using the Fisher's exact test. Benjamini–Hochberg correction was applied for multiple testing[72].

**Partial least squares path analysis**. To explore the complex system of relationships between micronutrient supplementation, place of residence and the bacteria and eukaryotes at 12 and 24 months, we conducted partial least squares (PLS) path analysis using the plspm 0.4.9 package in R[74]. Microbial read counts were normalized using center-log transformation after removing taxa with less than 0.01% abundance across all samples. The path matrix (i.e., directional arrows) was designed to assess the interrelationship between supplementation, place of residence, and microbiomes at different timepoints, including (1) eukaryotes at 12 months, (2) eukaryotes at 24 months, (3) bacteria at 12 months and (4) bacteria at 24 months. Cross-correlations among factors were calculated using the plspm::plspm function with the following parameters: taxa blocks (representing eukaryote or bacterial OTU matrices at each time point) were set to the *PLS-score* mode, scaling for supplementation and place of residence was set to *Nominal*, all other parameters were at default. PLS is a supervised dimensionality reduction technique that parallels principal component analysis. It reduces each high dimensional OTU matrix into a latent variable (PLS-score) that summarizes the interlinked differences in microbial community patterns that are related to supplementation and place of residence. The strengths of these cross-correlations are indicated by path coefficients, which can conceptually be understood as correlation coefficients. To test the stability of the path coefficients, we additionally implemented a bootstrapping procedure within the plspm package, which iteratively recalculates all path coefficients on randomly resampled participants with replacement. We also applied the plspm.groups function to test for differences in path coefficients between nutritional groups.

All microbial data and statistical analyses were carried out with R version 4.0.2[75].

**Ethics approval**. All clinical data and stool samples used for analysis were collected in a previously conducted cRCT (NCT00705445)[27] that was approved by the Ethics Review Committee (ERC) of AKU (752-Peds/ERC-07). All participants enrolled in the parent cRCT (NCT00705445) provided written informed consent for the routine collection of stool specimens for analysis of their intestinal microbial communities. Ethical approval for the retrospective analysis of clinical data and stored stool samples was granted by the research ethics board (REB) at SickKids, Toronto (REB No. 1000054244), the ERC at AKU, Karachi, Pakistan (4840-Ped-ERC-17), and the National Bioethics Committee (NBC) Pakistan (4-87/NBC-277/17/1191). A waiver of consent for this retrospective analysis was granted by each of the above-mentioned regulatory boards.

**Reporting Summary**. Further information on research design is available in the Nature Research Reporting Summary linked to this article.

## Data availability
The 16 S rRNA and 18 S rRNA amplicon data generated in this study have been deposited in the NCBI Sequence Read Archive database under the accession code PRJNA717317. The OTU count data generated in this study are provided in Supplementary Data 2 and Supplementary Data 5. The Ribosomal Database Project version 16 [https://sourceforge.net/projects/rdp-classifier/], SILVA v132 non-redundant reference database [https://www.arb-silva.de/download/archive/] and NCBI nucleotide database [https://www.ncbi.nlm.nih.gov/nucleotide/] used for taxonomic classification in this study are available under the provided weblinks.

## Code availability
R code for analyses is available in the GitHub repository ParkinsonLab/gut-eukaryotes-malnutrition-and-micronutrient-supplementation[76] [https://doi.org/10.5281/zenodo.5606073].

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

## Acknowledgements

We thank Imran Ahmed (Aga Khan University, Karachi, Pakistan) and Didar Alam (Aga Khan University) for assistance with organizing stool sample shipments from Pakistan to Canada and providing secure access to clinical data. We also appreciate the helpful advice and insights from Amel Taibi and Elena Comelli in addressing challenges encountered during the extraction of sample DNA. This work was supported by an HSBC Bank Canada Catalyst Research Grant from the Hospital for Sick Children awarded to C.B., R.H.J.B., D.S.G., J.P., and L.G.P.; the Canadian Institute for Health Research grant PJT-152921 to J.P.; Restracomp scholarship administered by the Research Training Centre (Hospital for Sick Children) and a graduate scholarship from the Government of Ontario to A.P. Computing resources were provided by the SciNet High Performance Computing (HPC) Consortium; SciNet is funded by the Canada Foundation for Innovation under the auspices of Compute Canada, the Government of Ontario, Ontario Research Fund - Research Excellence, and the University of Toronto.

## Author contributions

L.G.P., Z.A.B., J.P., and R.H.J.B. conceived and designed the study. S.S. and Z.A.B. participated in the original collection of clinical samples. A.P. isolated DNA and processed the sequencing data. P.W.W. and D.S.G. aided in the design of amplicon generation. A.P. and C.B. analyzed the data and wrote the paper and all authors reviewed and/or edited the paper.

## Competing interests

The authors declare no competing interests.
