## [Peer Review File · Nature Communications]

REVIEWER COMMENTS

Reviewer #1 (Remarks to the Author):

Title: Micronutrient supplements with iron promote disruptive protozoan and fungal communities in the developing infant gut (NCOMMS-21-24468)

Summary: This paper sought to examine the impact of micronutrient supplementation on gut microbiota, including bacteria, fungi, protozoa, and helminths among a subset of 12- and 24-month-old children (n=80) participating in an RCT.

Reviewer's Comments:

Key results: This research draws from the existing research on nutrition and the gut bacteria to examine an understudied area in microbiome research: the impact of micronutrients on eukaryotic microbiota. This is important as eukaryotes are often overlooked in microbiome research despite being part of the overall microbial community and likely having key roles in homeostasis, tolerance, and disease resistance. Key strengths include paired stool samples for all children (n=80) along with complete WLZ data, comparison between urban and rural areas, and an under-explored age group (2nd year of life). This study found that micronutrient supplements and zinc were associated with limited growth of protozoan and fungal taxa, compared to micronutrients and no zinc.

Validity:

1. This manuscript did not have and major flaws prohibiting its publication.

Originality & significance:

1. The results presented are of interest to researchers in nutrition and gut microbiome.

Data & methodology:

1. In addition to publishing the raw sequences, including a description of how the code was reproduced successfully by a 3rd party and availability of this code and dataset would strengthen the results, considering the speed at which gut microbiome analysis pipelines update and change. For example, the authors could import their 16S data into Qiita (www.qiita.ucsd.edu) and have data replicated by a service at their data institutions.

2. We were not able to locate or access BioProject PRJNA717317 in the NCBI Sequence Read Archive.

3. Please consider describing the stool sample collection process (materials, instructions/steps/protocols used); these are not described in detail for replication here or in the trial publication (Soofi et al. 2013).

Appropriate use of statistics and treatment of uncertainties:

1. Statistical tests were appropriate and included multiple comparisons testing.

2. In Line 519-520, please give a rationale for the rarefaction levels chosen.

Suggested improvements:

1. Please consider adding the zinc piece to a revised manuscript title; currently, the title suggests that the difference between arms was iron, but this doesn't seem to be the case.

2. It may be useful to use a reporting guideline such as CONSORT or appropriate version for secondary analyses of trials to ensure all necessary information is included.

3. It will be helpful, in an additional supplementary table, to compare the age, sex, WLZ and other characteristics of the 671, 646, and 659 in the control, MNP, and MNP + zinc groups in the trial at 24 months of age, to the 24, 29, and 27 infants, respectively, included in this ancillary study. If characteristics are similarly distributed, this will allow increased generalizability to the parent trial among those who were within the WLZ criteria, did not use antibiotics, and did not have diarrhea.

4. Please consider using the phrase "participant(s)" rather than "subject(s)".

5. Please include whether parents consented to this sub-study and/or to the stool sample collections as part of the parent trial.

6. Line 493-495: Please consider defining "nutritional status" (in the list of explanatory variables for the model explaining diversity).

7. Table 1:

a. While some factors of the microbiome including initiation of breastfeeding and premature birth are described in Table 1, type of birth (vaginal vs. Caesarean) is not included, which is another influencer of the infant gut microbiome, and could be added if this information is available.

b. Further to "initiation of breastfeeding", as nearly 100% of the infants initiated breastfeeding prior to the study, it may be interesting to compare the microbiome of infants who had exclusive breastfeeding for the first 6 months of life, vs. partial breastfeeding/early introduction of complementary foods, and whether this affects the impacts of MNP ± zinc or control.

References:

1. A recent review on complementary feeding, undernutrition and the gut microbiome was published this past June (Chehab et al. 2021). Chehab et al. found no studies on eukaryotic organisms in this context and therefore the review may be worth citing as further support/rationale for the current article.

2. While most methods (algorithms, pipelines) are cited, the Benjamini-Hochberg Correction is missing a reference.

Clarity and context:

1. Please consider adding the age group to the abstract. For example, in Line 29, the authors can add "...collected from children [insert: at 12 and 24 months of age]..."

2. Line 30-33, regarding the zinc findings, it should be clear that zinc supplementation is not being recommended for modifying the gut microbiome, particularly given the sample sizes of 24, 29, and 27 infants which, further divided by WLZ, include only 7 to 20 participants per group. Please add that there is more work needed to understand clinical significance and impact on functional and other health outcomes.

Please indicate any particular part of the manuscript, data, or analyses that you feel is outside the scope of your expertise, or that you were unable to assess fully.

Please address any other specific question asked by the editor via email.

Decision:

B. Invite the authors to revise their manuscript to address specific concerns before a final decision is reached

Reviewer #2 (Remarks to the Author):

Popovic et al: "Micronutrient supplements with iron promote disruptive protozoan and fungal communities in the developing infant gut"

General Comments:

Popovic et al tackle a challenging task in the microbiota field at multiple levels – 1) systems-level microbiota analysis in the context of a human clinical trial and 2) relating the oft-studied human bacterial microbiota and the less-addressed eukaryotic microbiota. As efforts to address malnutrition around the world increasingly consider the (positive and negative) effects of nutritional supplements on the gut microbiota as a mediator of those supplements' effect on the host, work like that described here is key to both bolster and temper enthusiasm over strategies based in part of considerations of the microbiota. The observations here are important, however, the manuscript would benefit significantly from an increased focus and clarity of presentation of results. The manuscript begins with a focus on the effects of MNP supplementation, but strays more into off-topic (though interesting) extraneous effects on the eukaryotic microbiota. It is difficult to follow the narrative regarding treatment, vs geography, vs time.

Specific Comments:

Line 1: Title – “disruptive” is not well-supported throughout. The specific role of the eukaryotic microbiota and what is disruptive about it doesn’t come through the text.

Line 58: (and throughout) “Prevalence.” Common support for focusing on the bacterial microbiota rather than the eukaryotic microbiota involves the relative abundance of bacteria to eukaryotes in the gut. Was any spike-in analysis applied to sequencing to determine absolute abundance? If not, this should be acknowledged.

Line 89: A simple schematic of the trial would be a more effective queue here for the reader, as the additional analysis conducted in this manuscript was clearly not the focus of the “original” cRCT trial. If this is not true, there should be a more significant focus on the clinical outcomes of the trial. Please clarify.

Line 89-91: WLZ scores here could include a broad range (MAM, SAM, and healthy). Would be helpful to mention in main text the range of WLZ scores included in the “WLZ” group. Also suggest to refer to this group consistently as “undernourished” rather than “WLZ”.

Line 101: Classification, however broad, is an important point. What were the “other” unclassified eukaryotic OTUs? How were they classified?

Line 123: Statistical methods for these analyses are not clearly indicated, and should be introduced here.

Line 130: Is this a consideration of age x treatment interaction, or just age?

Line 208: Define phylogenetic variance.

Lines 214-220: Are these significant results? Significance reporting in the later portions of the paper is lacking.

Line 222-on: Network analyses only reflect data from 1-2 timepoints. Is this enough longitudinal data to justify the conclusions made? This point should be discussed, considering the strategies of cross-individual analysis described.

Line 224: “Interactions, calculated as edges per node...” The network analyses are interesting, but need to be focused and written to give the reader more intuition as to what “edges” and interactions mean.

Lines 260-on: This section is hard to interpret and feels off-topic (CTL vs MNP vs MNP + Zn). Is there an easier tie to the original MNP supplementation? Also, there is a considerable reliance on nonsignificant statistical results in the later portions of the paper. One path to focus might be to rely further on significant tests first.

Line 294: “indirect association” is not justified. How specifically is the effect of eukaryotes on prokaryotes indirect, given the results of Fig 5?

Lines 320: What is the GEMS study? First time described is in the discussion.

Line 420: “not supported by robust bootstrapping” this is an important consideration. Should the reader care if the findings aren’t supported by the more robust methods? Why weren’t the more robust methods applied first?

Line 533-547: Methods description, particularly of PLS analysis, is extremely jargon-filled, and not very informative. I don’t think a reader would be able to replicate results using this description.

Figures:

Fig 1. Was the trial performed for the purpose of microbiota analysis? If not, I don’t think the WLZ results should lead. Plots are very hard to parse color-wise (this is a general comment throughout the figs). Intervals lines in 1b. are over-plotted and make following the actual trends tough to follow.

Fig 2. 2c – what’s the purpose of these rarefactions? No significance is reported. Should be supplementary. This figure in general is extremely difficult to parse color-wise.

Fig 3. 3c – why chose MDS vs PCA is earlier fig. Should be justified.

Fig 4. Significant comparisons are rarely indicated. Hard to tell which results a reader should pay attention to. 4e networks are interesting, but the importance of bordered circles vs unbordered is unclear. Should justify (somewhere) how consideration of different numbers of samples in different network reconstructions can affect results.

Fig 5. How is the effect of supplementation on bacteria indirect via eukaryotes based on this fig? There is a strong significant interaction between supplementation and bacteria, at least at 12mo. I don’t understand the “indirect” claims given this figure. Also, the focus in the text on this observation seem off-topic, as the manuscript is ostensibly not about MNP effects on bacteria at all.

REVIEWERS COMMENTS

Reviewer #1

Key results: This research draws from the existing research on nutrition and the gut bacteria to examine an understudied area in microbiome research: the impact of micronutrients on eukaryotic microbiota. This is important as eukaryotes are often overlooked in microbiome research despite being part of the overall microbial community and likely having key roles in homeostasis, tolerance, and disease resistance. Key strengths include paired stool samples for all children (n=80) along with complete WLZ data, comparison between urban and rural areas, and an under-explored age group (2nd year of life). This study found that micronutrient supplements and zinc were associated with limited growth of protozoan and fungal taxa, compared to micronutrients and no zinc.

Response: We would like to thank the reviewer for their time in careful reviewing the manuscript and the constructive comments they have provided.

Validity:

1. This manuscript did not have and major flaws prohibiting its publication.

Originality & significance:

1. The results presented are of interest to researchers in nutrition and gut microbiome.

Data & methodology:

1. In addition to publishing the raw sequences, including a description of how the code was reproduced successfully by a 3rd party and availability of this code and dataset would strengthen the results, considering the speed at which gut microbiome analysis pipelines update and change. For example, the authors could import their 16S data into Qiita (www.qiita.ucsd.edu) and have data replicated by a service at their data institutions.

Response: We agree that reproducibility is key for microbiome analyses. In response to this comment we now make all scripts used in the analyses available in a dedicated github repository (<https://github.com/ParkinsonLab/gut-eukaryotes-malnutrition-and-micronutrient-supplementation>). We further provide details on the relevant R software package versions in the methods section. In addition, we would be happy to work with the editor to fulfill any additional journal requirements regarding the use of third-party services to replicate the data and code.

2. We were not able to locate or access BioProject PRJNA717317 in the NCBI Sequence Read Archive.

Response: We apologize for neglecting to provide the reviewer link to the sequence data in the previous version. We had originally set the data for public release on Sept 1, 2021, and the data is currently accessible at: <https://www.ncbi.nlm.nih.gov/bioproject/PRJNA717317>

3. Please consider describing the stool sample collection process (materials, instructions/steps/protocols used); these are not described in detail for replication here or in the trial publication (Soofi et al. 2013).

Response: As requested we now provide additional details on stool collection in Methods (Pages 23 and 24):

Routine stool samples were collected from all participants at enrollment, 6, 12, 18, and 24 months of age. Caregivers were provided with a stool collection kit (plastic diaper, stool collection container, spatula, and gloves) and training on how to self-collect stool samples, document the time of defecation, and notify study staff that a sample has been collected. Stool samples were picked up at participant's homes by study staff within 1 hour of defecation and immediately transported to the field research office in a 2-4°C cold box. Prior to pick-up, stool samples were kept at room temperature in a shaded area of the participant's household. Up to eight aliquots (approximately 800 mg to 1 g per aliquot) were generated from each stool sample, of which 5 to 6 were marked for future analyses. All aliquots were transported to Aga Khan University (AKU) within 4-6 hours of preparation and aliquots targeted for future analyses were immediately frozen at -80°C, without any additives. Stool samples analyzed in this study were shipped from AKU to the Hospital for Sick Children (SickKids) on dry ice where they were stored at -80°C until analysis.

Appropriate use of statistics and treatment of uncertainties:

1. Statistical tests were appropriate and included multiple comparisons testing.

2. **In Line 519-520**, please give a rationale for the rarefaction levels chosen.

Response: Rarefaction levels (25,000 reads for bacteria and 1,000 reads for eukaryotes) were the same as those used for significance testing of alpha diversity, unifrac compositional differences for eukaryotes, as well as the network analyses. Selection of read depths was based on balancing:

(1) sufficient sampling of OTUs. For bacteria, a threshold was chosen beyond the “elbow” of the rarefaction curve, where it begins to flatten.

(2) retaining maximum sample representation (power), due to low sample numbers in certain groups (e.g. 14 undernourished children receiving MNP+Zinc), and highly variable read depths in eukaryote data.

For example, at a eukaryote rarefaction depth of 1000, 11 samples are removed from analysis due to low counts, while at a depth of 5000, 27 samples are removed. Additional text explaining this justification has been added to Methods (Page 26), and a supplementary figure has been included showing numbers of samples within each group available for analysis at each read depth (Supplementary Figure 6).

Suggested improvements:

1. Please consider adding the zinc piece to a revised manuscript title; currently, the title suggests that the difference between arms was iron, but this doesn't seem to be the case.

Response: We agree with the reviewer is correct that in addition to iron and/or vitamins, zinc has an impact on the microbiome. We have therefore altered the title in line with this recommendation to:

“Micronutrient supplements can promote disruptive protozoan and fungal communities in the developing infant gut”

2. It may be useful to use a reporting guideline such as CONSORT or appropriate version for secondary analyses of trials to ensure all necessary information is included.

Response: This is an excellent suggestion. In the absence of an appropriate version for secondary analyses of trials, we used the CONSORT 2010 checklist, addressing the appropriate items. This checklist is provided in supplementary information and referred to in the Methods (page 24).

3. It will be helpful, in an additional supplementary table, to compare the age, sex, WLZ and other characteristics of the 671, 646, and 659 in the control, MNP, and MNP + zinc groups in the trial at 24 months of age, to the 24, 29, and 27 infants, respectively, included in this ancillary study. If characteristics are similarly distributed, this will allow increased generalizability to the parent trial among those who were within the WLZ criteria, did not use antibiotics, and did not have diarrhea.

Response: As requested, we now provide participant characteristics from the parent trial as well as the microbiome substudy, grouped by supplementation arm, as Supplementary Table 1.

4. Please consider using the phrase “participant(s)” rather than “subject(s)”.

Response: We appreciate the suggestion and have updated the manuscript accordingly.

5. Please include whether parents consented to this sub-study and/or to the stool sample collections as part of the parent trial.

Response: All stool samples for microbiome analysis were collected in a cluster randomized controlled trial (cRCT) (NCT00705445) that was conducted between November 2008 and December 2011 (Soofi *et al.* 2013. Lancet) and approved by the Ethics Review Committee (ERC) of Aga Khan University (752-Ped/ERC-07). All participants enrolled in the cRCT (NCT00705445) provided written informed consent for the routine collection of stool specimens for analysis of their intestinal microbial communities. In this study, new lab analyses were performed on a sub-set of these previously collected and stored stool samples. No new biological samples or clinical data were collected as part of this study. Ethical approval for the retrospective analysis of stored stool samples was granted by the National Bioethics Committee (NBC) of Pakistan (NBC-277), the ERC at Aga Khan University (4840-Ped-ERC-17), and the Research Ethics Board (REB) at the Hospital for Sick Children (REB No. 1000054244). Given that participants in the cRCT provided written informed consent for the analysis of their intestinal microbial communities from stool samples and that unlikely that participants from a cRCT conducted between 2008 and 2011 could be traced for the collection of a second consent form, a waiver of consent was granted by NBC, ERC, and REB. Text outlining the above has now been added to the “Ethics Approval” section of the manuscript (Page 30).

6. Line 493-495: Please consider defining “nutritional status” (in the list of explanatory variables for the model explaining diversity).

Response: Nutritional status has now been defined as per the suggestion (Page 27).

7. Table 1:

a. While some factors of the microbiome including initiation of breastfeeding and premature birth are described in Table 1, type of birth (vaginal vs. Caesarean) is not included, which is another influencer of the infant gut microbiome, and could be added if this information is available.

Response: We agree with the reviewer that this would be an excellent factor to include. Unfortunately we do not have access to this data. However, it should be noted that delivery by Caesarean in these localities is uncommon.

b. Further to “initiation of breastfeeding”, as nearly 100% of the infants initiated breastfeeding prior to the study, it may be interesting to compare the microbiome of infants who had exclusive breastfeeding for the first 6 months of life, vs. partial breastfeeding/early introduction of complementary foods, and whether this affects the impacts of MNP ± zinc or control.

Response: We concur with the reviewer that this would be an interesting facet to add to the study and in fact prior to this secondary study commencement we investigated if this could be addressed. Unfortunately, the original study was not designed to allow for this analysis and we found that most mothers, i.e., 78 out of 80 (97.5%) reported partial breastfeeding already at 6 months. Thus, with our current study sample size, we were not able to address this question.

However, given the reviewers comment and likely general interest in this question, we decided to add to the table of characteristics (Table 1) the breakdown of breastfeeding status (exclusive breastfeeding, partial breastfeeding, no breastfeeding) reported by mothers at 6 months.

References:

1. A recent review on complementary feeding, undernutrition and the gut microbiome was published this past June (Chehab et al. 2021). Chehab et al. found no studies on eukaryotic organisms in this context and therefore the review may be worth citing as further support/rationale for the current article.

Response: We would like to thank the reviewer for highlighting this study and now include it in the introduction to lend further support for the current study (Page 4).

2. While most methods (algorithms, pipelines) are cited, the Benjamini-Hochberg Correction is missing a reference.

Response: We apologize for the oversight and now include the citation (Page 27).

Clarity and context:

1. Please consider adding the age group to the abstract. For example, in Line 29, the authors can add “...collected from children [insert: at 12 and 24 months of age]...”

Response: We have added information on the ages to the abstract, as per reviewer suggestion.

2. **Line 30-33**, regarding the zinc findings, it should be clear that zinc supplementation is not being recommended for modifying the gut microbiome, particularly given the sample sizes of 24, 29, and 27 infants which, further divided by WLZ, include only 7 to 20 participants per group. Please add that there is more work needed to understand clinical significance and impact on functional and other health outcomes.

Response: We concur with the reviewers sentiment and have added a statement at the end of the abstract to emphasize that more work needs to be done to evaluate the clinical significance and impact on health outcomes in line with the recommendation (Page 2).

Reviewer #2 (Remarks to the Author):

Popovic et al: “Micronutrient supplements with iron promote disruptive protozoan and fungal communities in the developing infant gut”

General Comments:

Popovic et al tackle a challenging task in the microbiota field at multiple levels – 1) systems-level microbiota analysis in the context of a human clinical trial and 2) relating the oft-studied human bacterial microbiota and the less-addressed eukaryotic microbiota. As efforts to address malnutrition around the world increasingly consider the (positive and negative) effects of nutritional supplements on the gut microbiota as a mediator of those supplements’ effect on the host, work like that described here is key to both bolster and temper enthusiasm over strategies based in part of considerations of the microbiota. The observations here are important, however, the manuscript would benefit significantly from an increased focus and clarity of presentation of results. The manuscript begins with a focus on the effects of MNP supplementation, but strays more into off-topic (though interesting) extraneous effects on the eukaryotic microbiota. It is difficult to follow the narrative regarding treatment, vs geography, vs time.

Response: We would like to thank the reviewer for their support of this work. At the same time we appreciate the concern regarding the narrative. To address this concern we have edited the results to explain the rationale for each section to make the narrative easier to follow.

Specific Comments:

Line 1: Title – “disruptive” is not well-supported throughout. The specific role of the eukaryotic microbiota and what is disruptive about it doesn’t come through the text.

Response: This is a fair comment and we acknowledge the need to emphasize more in the text, the disruptive influence of the eukaryotic microbiota. In using the term ‘disruptive’, we refer to the potential for supplements to promote growth of taxa associated with “disruptive” effects on the gut environment or more broadly on host health (e.g. diarrhea and inflammation, invasive infection). For example, MNPs are associated with increased carriage of mucormycete fungi, which can cause invasive infections in malnourished infants², and *Toxoplasma*, known to cause inflammation and promote gut Proteobacterial growth during acute infection³. More generally, they are associated with an overall change in protozoan composition and interactions, as well as a reduction in bacterial diversity. Our integrated PLS model suggests that these changes in eukaryotes at 12 months may influence (or disrupt) the make-up of bacterial communities.

We have added commentary within the introduction, results, discussion and conclusion to help make this more apparent (Pages 4, 14, 18-20 and 22).

Line 58: (and throughout) “Prevalence.” Common support for focusing on the bacterial microbiota rather than the eukaryotic microbiota involves the relative abundance of bacteria to eukaryotes in the gut. Was any spike-in analysis applied to sequencing to determine absolute abundance? If not, this should be acknowledged.

Response: No spike-in analysis was carried out in this study. We agree with the reviewer that determining the relative abundance of bacteria to eukaryotes would be informative, particularly in ascertaining how MNP supplements impact eukaryote growth. We have added a statement to acknowledge this limitation in the discussion, and its potential utility in future studies (Page 21).

Line 89: A simple schematic of the trial would be a more effective queue here for the reader, as the additional analysis conducted in this manuscript was clearly not the focus of the “original” cRCT trial. If this is not true, there should be a more significant focus on the clinical outcomes of the trial. Please clarify.

Response: We apologize for the lack of clarity. The reviewer is correct that the focus of the parent cRCT trial was on the growth and morbidity of children in response to MNPs. We have included a simple schematic in Figure 1a (please also see below) depicting the set up of the parent trial, as well as the sub-selection of children for microbiome profiling in this substudy. We believe this provides an informative introduction to our study.

Line 89-91: WLZ scores here could include a broad range (MAM, SAM, and healthy). Would be helpful to mention in main text the range of WLZ scores included in the “WLZ” group. Also suggest to refer to this group consistently as “undernourished” rather than “WLZ”.

Response: In response to this suggestion, we have added the 95% CI WLZ range for each group in the text. The undernourished group was defined as weight-for-length <-2 and is referred to as “undernourished” throughout. We refer to the comparator group, defined as WLZ >-1 , as “reference WLZ” (Page 5).

Line 101: Classification, however broad, is an important point. What were the “other” unclassified eukaryotic OTUs? How were they classified?

Response: We agree! And in response to this comment we now provide more detail to the text, with information on the numbers of high quality clustered reads after filtering, and how unassigned or other eukaryote reads were classified. Of the 11,639,233 paired reads that were produced by sequencing, 7,167,464 were high quality and clustered into OTUs, and 4,386,494 were classified as microbial eukaryote. The remaining reads were assigned primarily as plant (embryophyta, 1,600,136 reads), mammalian (most likely host, 312,235 reads), arthropod (211,284 reads), contaminating bacterial

(322,725 reads, not predicted as rRNA) or remained unassigned (433,435 reads). Taxonomic classification was carried out using a stepwise approach: (1) the SINA aligner with the SILVA v132 non-redundant reference database (<https://www.arb-silva.de/aligner/>); (2) SINTAX (within USEARCH) with the same SILVA v132 non-redundant reference database; and finally (3) remaining unclassified sequences were compared using BLAST to the NCBI non-redundant nucleotide database with minimum cut-offs of 90% query coverage and percent identity. The reason for using this stepwise approach is that the SINA aligner, for example, does not identify *Entamoeba* 18S sequences, but we can identify them using SINTAX or BLAST. With BLAST, we identified predominantly contaminating bacterial or human/mammalian sequences that were not derived from the rRNA locus. Text describing the breakdown of ‘other’ unclassified eukaryotic OTUs has been added (Page 6). Details on the classification scheme are provided in Methods (Page 26).

Line 123: Statistical methods for these analyses are not clearly indicated, and should be introduced here.

Response: We apologize for the confusion. We implemented generalized linear models to test for associations between variables and alpha diversity, and we applied Fisher’s exact tests to compare eukaryote carriage (prevalence) among groups. We have added this to the text (Page 7).

Line 130: Is this a consideration of age x treatment interaction, or just age?

Response: Here we only calculated differences in eukaryote carriage (prevalence) using the Fisher’s exact test between rural and urban sites, first including both age groups, then separately in 12 month-old and 24 month-old children. As sample numbers were low, we did not further subdivide by treatment. We did, however, include all four explanatory variables (age, location of residence, treatment and nutritional status) in our linear models when testing for associations with alpha diversity, and in our adonis tests for associations with beta diversity.

Line 208: Define phylogenetic variance.

Response: By phylogenetic variance we were referring to the sample dispersions, or distances to the group centroid, based on weighted Unifrac dissimilarities which incorporate phylogenetic “relatedness” of taxa. We have added a clarification to the text (Page 11).

Lines 214-220: Are these significant results? Significance reporting in the later portions of the paper is lacking.

Response: Yes, all results reported from differential abundance testing (using DESeq2) are significant. P-values range from $p < 0.001$ to $p < 0.05$. We have added a clarification and p-values to the text (Page 11), and have indicated significant results in the boxplots in Fig 3g. We also note that there was an error in the y-axis labelling of Fig 3g which has now been corrected.

We further agree with the reviewer with respect to significance reporting in the latter portion of the paper and have now included significance testing for Fig 4a and Fig 4b, to strengthen the microbial interaction network analysis (Pages 12-14).

Line 222-on: Network analyses only reflect data from 1-2 timepoints. Is this enough longitudinal data to justify the conclusions made? This point should be discussed, considering the strategies of cross-individual analysis described.

Response: The reviewer is correct that inclusion of multiple timepoints would provide better support for conclusions about network complexity during growth. We have replaced words in this section of the text (Page 12) that imply development over time with terms which simply compare two variable states (e.g. 'increased' to 'were significantly higher', 'change' to 'difference'). We elected to retain the observation that our data, showing greater numbers of interactions in children at 24 months compared to 12 months, are consistent with the development of a more complex microbiome over time, previously shown by Stewart *et al.* (2018)⁴, as we believe it provides supporting context. We have, however, added this citation to make clear that this is not a novel conclusion drawn from observations of two timepoints, rather that our data is in line with previous findings. We have also noted at the end of this section (Page 14) that data from additional timepoints are needed to confirm whether the potentially destabilizing effects persist throughout the intervention, and whether any recovery of interactions is seen upon cessation of supplementation.

Line 224: "Interactions, calculated as edges per node..." The network analyses are interesting, but need to be focused and written to give the reader more intuition as to what "edges" and interactions mean.

Response: In response to this comment we have expanded this section to more clearly communicate the intent of the analysis, the meaning of 'interactions' in this context and those network properties used for comparisons, and have attempted to better guide the reader through the analyses (Page 12).

Lines 260-on: This section is hard to interpret and feels off-topic (CTL vs MNP vs MNP + Zn). Is there an easier tie to the original MNP supplementation? Also, there is a considerable reliance on nonsignificant statistical results in the later portions of the paper. One path to focus might be to rely further on significant tests first.

Response: We agree with the reviewer that the current presentation can be improved to better tie into the context of the main findings. We have thus changed the section sub-title and reorganized the paragraph in a manner that better maintains focus on the most relevant aspect of the study (i.e., influence of supplementation on microbial profiles). Furthermore, the text was edited regarding the results linked to place of residence, which we believe has significantly improved the focus of this section (Pages 14-16).

Line 294: "indirect association" is not justified. How specifically is the effect of eukaryotes on prokaryotes indirect, given the results of Fig 5?

Response: We agree that the sentence highlighted was unclear and given the previous comment, we are of the opinion that the focus should remain on the possible carryover effect of supplementation on microbiomes at 24 months (seen in both bacteria and eukaryotes) via its earlier effect at 12 months. It is to be noted that the correlations (direct arrows in Figure 5) between microbiomes at age 12 and 24 months are composed of upstream effects that can act in opposing directions (e.g. site and supplementation). Thus, the interpretation of total, direct and indirect effects of elements are specifically presented in the text and detailed in Supplementary Table 6. We have added in both the text and figure legend clarifications regarding what is meant by “indirect effects”, while trying to maintain a simplified overview (Pages 14-15). The text has also been revised with a more general reflection that our results justify the need to further investigate the effects of supplementation in vulnerable children living in different LMIC settings.

Lines 320: What is the GEMS study? First time described is in the discussion.

Response: We apologize for this oversight. The Global Enteric Multicenter Study (GEMS) is the largest and most comprehensive study of childhood diarrhea to date, with 22,568 children enrolled across seven developing countries in Asia and Africa⁵. We have added this description to the discussion (Page 17).

Line 420: “not supported by robust bootstrapping” this is an important consideration. Should the reader care if the findings aren’t supported by the more robust methods? Why weren’t the more robust methods applied first?

Response: Thank you for this comment. Prior to submission, we gave this much thought. Some studies do not use bootstrapping or report these validation parameters for PLS-path models, these results are not typically produced by default. However, we opted to include this test as we believe that this procedure does give an indication of stability that is not otherwise available. PLS can be sensitive to outliers which we carefully evaluated, but large standard errors on bootstrap can also be linked to resampling subsets of participants with unrecognized heterogeneity in their profiles, thus creating jumps in estimated coefficients. We know that the eukaryotic profiles are heterogeneous and with our sample size and the stochastic aspect of absence/presence of carriage in children, the random re-selection of participants performed with bootstrapping could exclude children that carry the eukaryotes with the strongest effects.

Thus, we think that the standard error on bootstrap reflects both heterogeneous carriage and responses, as well as possible unevaluated interactions. This would not be surprising. However, given our sample size, and stratification, we recognize that we are unable to tease this out. We chose to present the overview model since the results lined-up with our other findings, and as it effectively highlights our main message that: our understanding of the effects of supplementation needs to be further studied especially in vulnerable populations living in LMICs who encounter more diverse organisms, which may respond to nutrient bioavailability in ways not yet understood. We think that this model, including the standard error on bootstrap, helps to achieve this message.

In light of the above, we have added additional text explaining the bootstrapping approach and interpretation in Methods (Page 29), Results (Page 15) and Discussion (Page 22).

Line 533-547: Methods description, particularly of PLS analysis, is extremely jargon-filled, and not very informative. I don't think a reader would be able to replicate results using this description.

Response: We agree - upon re-reading this section we also felt that this section requires reducing the jargon and clarifying the approaches. We have therefore re-written the description of the PLS-path analysis, including not only the package reference but also list the main functions and parameter settings that were used. This hopefully clarifies the analytical steps in a way that would better help readers understand how the analysis was conducted. We have also added additional pointers to the figure legend of the PLS model to help ease interpretation of what is meant by "indirect path" (Page 29).

We have furthermore included a more detailed description of microbial network analysis, including definitions of network properties used for comparison and the statistical tests applied, and clarified methods sections pertaining to diversity analyses, to help the reader (Pages 26-28). We provide R scripts used in the construction of the PLS model and microbial networks in github (<https://github.com/ParkinsonLab/gut-eukaryotes-malnutrition-and-micronutrient-supplementation>). This link is provided under 'Code availability' (Page 30).

Figures:

Fig 1. Was the trial performed for the purpose of microbiota analysis? If not, I don't think the WLZ results should lead. Plots are very hard to parse color-wise (this is a general comment throughout the figs). Interval lines in 1b. are over-plotted and make following the actual trends tough to follow.

Response: The objective of the original trial was to assess the impact of micronutrient powders (including vitamins and iron, or vitamins, iron and zinc) on growth and morbidity. We have modified the text in the results section to clarify this point, and in line with an earlier comment, we have added a schematic of the parent trial with information on our retrospective microbiome substudy in Figure 1a. Our intent in Figure 1 was to introduce the characteristics of the substudy cohort, and to place them in the context of the parent trial population.

We further agree with the reviewer, with respect to the Fig. 1b interval lines and colours and have modified the figure accordingly.

Fig 2. 2c – what's the purpose of these rarefactions? No significance is reported. Should be supplementary. This figure in general is extremely difficult to parse color-wise.

Response: We used rarefaction curves to show the differences (or the lack thereof) between sample groups more clearly, and reported the calculated significance at a single rarefaction depth (indicated in the figure by the dotted lines). In the case of eukaryotes, in particular, the read depth between samples was highly variable, and samples are lost at high read depths (please see graphs below for the number of available samples per protozoan (a-c) or fungal (d-f) 18S read depth in each sample group, when divided by nutritional status and supplementation (*left*), age (*middle*), and place of residence (*right*)). We

therefore elected to test for significance at the read depth of 1000, where we have higher sample representation in each grouping. We have added as Supplementary Figure 6 plots showing the relationship between read depth and sample numbers.

We have made the colours more bold in order to make sample groupings more obvious, and attempted to simplify several graphs (e.g. Fig 1b). We have also attempted to ensure that different colours within the same graphs can be differentiated using the Coblis colour blindness simulator (<https://www.color-blindness.com/coblis-color-blindness-simulator/>). Where possible, we have kept colours for the different variables consistent throughout the figures, in order to help the reader.

Fig 3. 3c – why chose MDS vs PCA is earlier fig. Should be justified.

Response: We carried out both PCoA and NMDS ordinations on the data (please see below for bacterial data). While both showed separation between 12 and 24 month samples, we felt that NMDS ordination maximized and therefore better captured the compositional variance among samples when projected onto the 2-dimensional plot. Consequently, correlations with changing OTU abundances which were projected onto the plot would be more accurate.

In the eukaryotic data (Fig 2e), ordination using NMDS did not produce a stable model (stress value >0.2, independently calculated models did not converge to a single solution), although four clusters were similarly visible. Thus, we had included the PCoA analysis. However, upon further consideration of the reviewer’s comment on the use of different approaches, we carried out additional analyses of eukaryotic data using NMDS, increasing dimensionality and the number of random starts, and have succeeded in producing a stable solution. We have updated Figure 2e and the related Figure 2f, and the text (Pages 8 and 27). We believe the consistency in the approach helps to improve the interpretability of results as well as the narrative.

Fig 4. Significant comparisons are rarely indicated. Hard to tell which results a reader should pay attention to. 4e networks are interesting, but the importance of bordered circles vs unbordered is unclear. Should justify (somewhere) how consideration of different numbers of samples in different network reconstructions can affect results.

Response: We would like to thank the reviewer for highlighting these concerns. In response we have updated Fig 4a and 4b to include statistical significance, and have elaborated on the measures and statistical tests used in the methods and results sections. The original analysis in 4a examined differences in microbial network density between sample groups (defined as ‘the total number of significant associations among taxa’, divided by ‘the total number of taxa present’). We have improved this analysis to compare distributions of the significant associations held by taxa between sample groups (i.e. ‘numbers of significant associations (degrees) held by each taxon in a network’). Wilcoxon rank-sum tests were used to calculate significant differences, with a Benjamini-Hochberg correction for multiple testing. The outcome is a significant increase in associated taxa, inferred as interactions, at 24 months compared to 12 months, as shown by the initial comparison. The Fisher’s exact test was used to evaluate differences in proportions of inter-kingdom interactions.

We have also updated the figure legend in 4e to define the relevance of bordered circles. Bordered circles indicate those taxa with the 10 highest “betweenness” scores (taxa coordinating the most microbial interactions).

With regards to the difference in the number of samples used for network reconstructions, we were unfortunately limited in the number of available samples. As the reviewer has suggested, differences in the numbers of samples used to generate networks have the potential to impact results, with smaller sample numbers associated with increased false positive associations and reduced recall of true associations⁶. We have added a caveat acknowledging this limitation in the results and discussion (Pages 14 and 21).

Fig 5. How is the effect of supplementation on bacteria indirect via eukaryotes based on this fig? There is a strong significant interaction between supplementation and bacteria, at least at 12mo. I don’t understand

the “indirect” claims given this figure. Also, the focus in the text on this observation seem off-topic, as the manuscript is ostensibly not about MNP effects on bacteria at all.

Response: Again we apologize for the confusion regarding the claims of indirect interactions. In responding to the reviewer’s previous concerns regarding these analyses, we hope that this has now been addressed. The indirect effects refer to the fact that supplementation may have carry-over effects on microbiomes at 24 months via its early impact at 12 months.

By convention, PLS-path figures present only direct cross-correlations between adjacent factors, which are formally tested for significance of their strength. However, these direct correlations (direct arrows) in the figure between the microbiomes at age 12 and 24 months include also the combination effects of supplementation and site on microbiomes at 12 months.

An advantage of PLS-path analysis is that it can parse out the estimated contributions of opposing factors in the model (i.e. the positive correlation of site and negative correlation of supplementation with microbiomes) while estimating the direct effects between connected factors and indirect effects that act downstream via an earlier element. The text now more clearly directs interested readers to the more detailed results in Supplementary Table 6, which include both direct and indirect correlation coefficients. We have also added in both the text and figure legend clarifications regarding the meaning of “indirect effects”.

References:

- 1 Soofi, S. *et al.* Effect of provision of daily zinc and iron with several micronutrients on growth and morbidity among young children in Pakistan: a cluster-randomised trial. *The Lancet* **382**, 29-40, doi:10.1016/S0140-6736(13)60437-7 (2013).
- 2 Francis, J. R., Villanueva, P., Bryant, P. & Blyth, C. C. Mucormycosis in Children: Review and Recommendations for Management. *J Pediatric Infect Dis Soc* **7**, 159-164, doi:10.1093/jpids/pix107 (2018).
- 3 Raetz, M. *et al.* Parasite-induced TH1 cells and intestinal dysbiosis cooperate in IFN-gamma-dependent elimination of Paneth cells. *Nat Immunol* **14**, 136-142, doi:10.1038/ni.2508 (2013).
- 4 Stewart, C. J. *et al.* Temporal development of the gut microbiome in early childhood from the TEDDY study. *Nature* **562**, 583-588, doi:10.1038/s41586-018-0617-x (2018).
- 5 Kotloff, K. L. *et al.* Burden and aetiology of diarrhoeal disease in infants and young children in developing countries (the Global Enteric Multicenter Study, GEMS): a prospective, case-control study. *Lancet* **382**, 209-222, doi:10.1016/S0140-6736(13)60844-2 (2013).
- 6 Kurtz, Z. D. *et al.* Sparse and compositionally robust inference of microbial ecological networks. *PLoS Comput Biol* **11**, e1004226, doi:10.1371/journal.pcbi.1004226 (2015).

REVIEWERS' COMMENTS

Reviewer #1 (Remarks to the Author):

Thank you for addressing our comments and incorporating the feedback. No further comments.

Reviewer #2 (Remarks to the Author):

I appreciate the authors efforts to improve their manuscript, and think it has significantly improved from the first version reviewed. They have addressed each of my concerns as acknowledged in their response.

Response to reviewer's comments

REVIEWERS' COMMENTS

Reviewer #1 (Remarks to the Author):

Thank you for addressing our comments and incorporating the feedback. No further comments.

Reviewer #2 (Remarks to the Author):

I appreciate the authors efforts to improve their manuscript, and think it has significantly improved from the first version reviewed. They have addressed each of my concerns as acknowledged in their response.

Response: We appreciate the feedback and would like to thank both reviewers for the careful and considered review of our manuscript.